# Hybrid photoacoustic and fast super-resolution ultrasound imaging

Shensheng Zhao[1,2,3], Jonathan Hartanto [4], Ritin Joseph[5], Cheng-Hsun Wu[6], Yang Zhao [1,3,7] & Yun-Sheng Chen [1,2,3,7,8] ✉

The combination of photoacoustic (PA) imaging and ultrasound localization microscopy (ULM) with microbubbles has great potential in various fields such as oncology, neuroscience, nephrology, and immunology. Here we developed an interleaved PA/fast ULM imaging technique that enables super-resolution vascular and physiological imaging in less than 2 seconds per frame in vivo. By using sparsity-constrained (SC) optimization, we accelerated the frame rate of ULM up to 37 times with synthetic data and 28 times with in vivo data. This allows for the development of a 3D dual imaging sequence with a commonly used linear array imaging system, without the need for complicated motion correction. Using the dual imaging scheme, we demonstrated two in vivo scenarios challenging to image with either technique alone: the visualization of a dye-labeled mouse lymph node showing nearby microvasculature, and a mouse kidney microangiography with tissue oxygenation. This technique offers a powerful tool for mapping tissue physiological conditions and tracking the contrast agent biodistribution non-invasively.

Dual-modality PA/ultrasound (US) imaging takes the merits of both imaging modalities[1–11]. Critical features of PA imaging include high-contrast blood vessel images without the need for imaging agents[6], blood oxygen levels[12], essential components of tissue such as lipid and collagen[3], and the distribution of disease-associated biomarkers with contrast agents[13–17]. US imaging, on the other hand, provides blood perfusion, tissue morphology, and elasticity[18]. PA and US share the same image recording system that enables the interleaved dual-contrast scanning for an improved spatial and temporal registration of two complementary imaging datasets[19,20]. Together, they provide structural, physiological, functional, and molecular information in one imaging system[21]. The dual has been widely used in preclinical biomedical studies and recently for various clinical applications, including breast imaging[22,23], melanoma detection[24], sentinel lymph node analysis[25], endoscopic applications[26], and brain imaging[27].

While the dual modality provides versatile imaging capabilities, the trade-off between spatial resolution and penetration hinders high-resolution imaging in deep tissue. This problem can be partially alleviated by the recently developed super-resolution localization imaging techniques, including US localization microscopy (ULM)[28] and PA localization imaging[29,30]. These localization techniques share a similar idea, but the latter is limited by the relatively weak PA signal and requires special contrast agents[29,30]. ULM breaks the acoustic diffraction limit and can achieve up to ten-fold improvement in spatial resolution compared to traditional US imaging in theory[31–33]. Adding super-resolution ULM vascular imaging to the already powerful PA/US would significantly enhance the imaging capabilities and provide additional information for many clinical applications.

One prerequisite of ULM is to use microbubbles at a concentration that separates the microbubbles by at least one US diffraction

[1]Department of Electrical and Computer Engineering, University of Illinois Urbana-Champaign, Urbana, IL, USA. [2]Beckman Institute for Advanced Science and Technology, University of Illinois Urbana-Champaign, Urbana, IL, USA. [3]Holonyak Micro and Nanotechnology Laboratory, University of Illinois Urbana-Champaign, Urbana, IL, USA. [4]Department of Chemical and Biomolecular Engineering, University of Illinois Urbana-Champaign, Urbana, IL, USA. [5]Department of Materials Science and Engineering, University of Illinois Urbana-Champaign, Urbana, IL, USA. [6]Verily Life Sciences, South San Francisco, CA, USA. [7]Department of Bioengineering, University of Illinois Urbana-Champaign, Urbana, IL, USA. [8]Department of Biomedical and Translational Sciences, Carle Illinois College of Medicine, University of Illinois Urbana-Champaign, Urbana, IL, USA. ✉e-mail: yunsheng@illinois.edu

distance to maintain the precision of the standard localization method, such as point spread function cross-correlation (PSF-CC)[28,32,34,35]. Typically, in ULM, dozens of isolated microbubbles can be identified from surrounding tissues and blood[36]. Under such conditions, it requires thousands of US frames to reconstruct one ULM frame[28,37]. Even with the recent ultrafast US imaging (>500 frames/sec)[38], the ULM acquisition time is still in tens to hundreds of seconds per frame[36]. On the other hand, the acquisition time of PA imaging is 10–100 Hz. Long data acquisition (DAQ) of ULM complicates the integration with PA imaging in real-time, resulting in the low temporal resolution of dual PA/ULM imaging.

To shorten the acquisition time of dual PA/ULM imaging, it is necessary to accelerate ULM. One approach to achieve this is using highly concentrated microbubbles, but it creates the issue of overlapped point spread functions in each frame, which invalidates the conventional microbubble localization method. Compressed sensing methods are proposed to localize high-density overlapped microbubbles[39,40] and accelerate ULM, generating a super-resolution velocity map. However, the reported compressed sensing localization methods require patch-stitch image processing, which is computationally demanding and may affect localization accuracy. While deep-learning-based localization is one powerful method, it demands large training datasets for different organs[41]. Sparsity-based super-resolution correlation imaging overcomes this problem without microbubble localization[42,43]. However, a super-resolution velocity map has not been demonstrated using the method.

Here, we demonstrate a fast interleaved PA/ULM imaging scheme accelerated by SC optimization using a fast shrinkage-thresholding algorithm (FISTA). Our SC-ULM method can adapt to much higher microbubble concentrations compared to PSF-CC. The results show an up-to-28-fold improvement in DAQ in vivo. The speed improvement enables the interleaved PA/ULM imaging, minimizing the motion artifacts and providing better spatial-temporal registration either in 3D scanning or time-lapse imaging. To showcase the imaging capabilities, we first demonstrate 3D dual-contrast images, which spatially co-register PA dye-labeled lymph node images and ULM vascular images, enabling the differentiation of the blood vessels within and outside the lymph node. We further show co-registered time-lapse images of super-resolved renal microvascular velocity distribution recorded by

our SC-ULM and the kidney tissue oxygenation level recorded by label-free PA imaging. Because of interleaved recording, we can spatially co-register dual PA and ULM kidney images over time. As a result, we can identify the relation of renal blood flow to hemoglobin oxygenation at the exact location over time.

## Results

### The interleaved imaging sequence for a dual PA/fast ULM imaging system

The imaging scheme of dual-modal PA and fast super-resolution US imaging is shown in Fig. 1. A 15 MHz linear array US transducer (VisualSonics, MS250) combined with a customized bifurcated fiber bundle is controlled by the Verasonics US system to transmit US waves and collect PA and US signals. A detailed description of the imaging system is in Methods. The interleaved imaging sequence is designed to acquire both PA and ultrafast US images (Fig. 1b). For each imaging cycle, the PA acquisition time is around 0.1–0.5 sec (with a frame rate of 10 Hz) depending on the number of laser wavelengths (each wavelength takes 0.1 sec), whereas the fast ULM acquisition time is 0.5–1.5 sec, depending on the application. The laser wavelength is tuned for each cycle to collect multi-wavelength PA signals. After image reconstruction, spectral unmixing[12,44] and sparsity recovery are applied to multi-wavelength PA images and a series of US images, respectively, to generate dual-modal PA/ULM images. The critical steps of the sparsity recovery process are shown in Fig. 1c. Briefly, microbubble signals are extracted from US images using clutter filtering. After that, microbubble localizations are estimated using SC optimization frame by frame, and every frame's final recovered microbubble positions are overlaid to generate a super-resolution US image.

### SC optimization method recovers 37 times more microbubbles than the conventional method in synthetic images

The temporal resolution of a dual-modality PA/ULM is limited by the DAQ time of ULM. To accelerate ULM, we used SC optimization to accelerate the dual imaging. We first evaluated the microbubble recovery ability of SC optimization in synthetic US microbubble images. We simulated 100 microbubbles that were randomly distributed in a $2.5 \times 2.5$ mm region with amplitudes following Gaussian

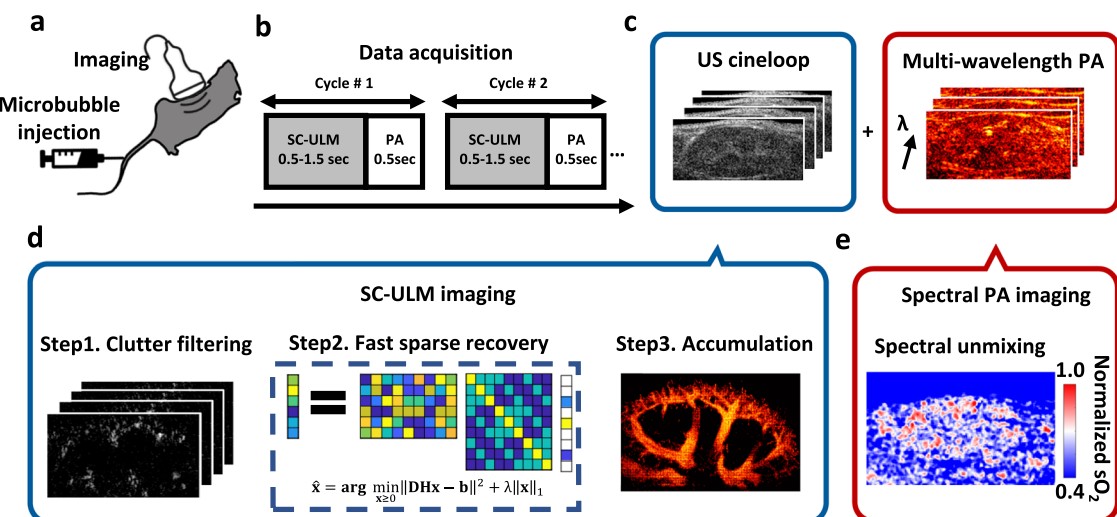

**Fig. 1 | Experimental concept of hybrid PA/fast ULM imaging.** The dual in vivo images are generated and processed with the following protocols: **a** Animals are injected with microbubble solution before (or during) imaging. **b** During the DAQ, multi-wavelength photoacoustic (PA) imaging and plane-wave fast ultrasound imaging are recorded alternately at each position. The minimum DAQ time for fast ultrasound localization microscopy (ULM) imaging is determined by the characterization time of an averaged pixel saturation curve. **c** Ultrasound (US) images and PA images of multiple wavelengths are stored and processed separately with procedures shown in (**d**) and (**e**). **d** Sparsity-constrained (SC)-ULM imaging procedure includes three main steps: clutter filtering, SC recovery over frames, and location accumulation. **e** Blood oxygen saturation (sO₂) PA images are generated through linear spectral unmixing from multi-wavelength PA images.

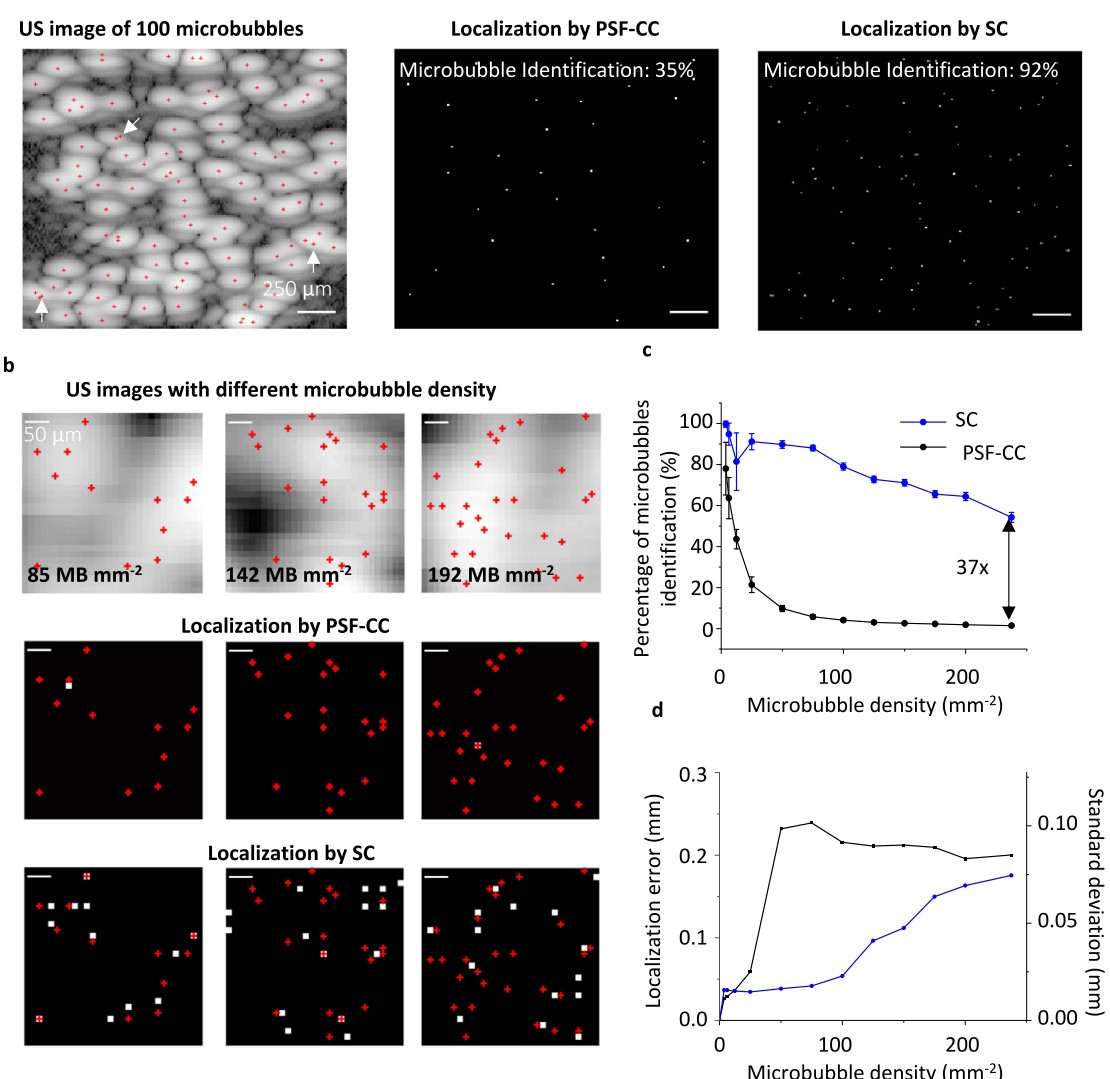

**Fig. 2 | Microbubble localization efficiency of SC-ULM and PSF-CC ULM with synthetic imaging data. a** Left to right: a simulated ultrasound image of 100 microbubbles randomly located in an area of 6.25 mm² (an average density of 16 microbubbles mm⁻²) with their ground-truth locations (marked as red dots), the recovered microbubble locations (white dots) using the PSF-CC method, and the recovered microbubble locations (white dots) using the SC method. The SC method recovers 92% of microbubbles, while PSF-CC recovers 35%. **b** Left to right columns represent the zoom-in images of three selected areas with different microbubble densities. The first row is the simulated microbubble ultrasound images. The second and the third row compare the recovered microbubble locations using the two methods. The white dots represent recovered microbubble locations, and the red dots represent the ground-truth locations. **c** The average microbubble recovery percentage as a function of microbubble densities using the SC method (blue) and the PSF-CC method (black). The error bar is the standard deviation ($N = 20$). **d** The average localization error (left y-axis) and the standard deviation ($N = 20$, right y-axis) as a function of microbubble densities using the two methods.

distribution using a US simulation module (Verasonics Simulator, and Supplementary Note 2). From Fig. 2a, the synthetic US image shows that although microbubbles are densely distributed (16 microbubbles mm⁻²) and overlapped, our SC algorithm identifies 92 out of 100 microbubbles and only a small portion of highly overlapped bubbles is beyond the resolution of the algorithm (such as bubbles indicated by the white arrows in Fig. 2a). As a comparison, the PSF-CC method barely identifies overlapped microbubbles and only recovers 35 out of 100 microbubbles of the same image. When increasing the microbubble density to 85, 142, and 192 microbubbles mm⁻², respectively, in Fig. 2b, we found that the sparsity algorithm remains capable of differentiating the microbubbles, although with sacrificing the localization accuracy. However, the PSF-CC method fails to identify the overlayed high-density microbubbles. A detailed discussion of the localization limitation of PSF-CC and SC methods is in Supplementary Note 1. We introduced two metrics for quantitative

analysis to characterize a localization event: the percentage of identified microbubbles and the localization precision as a function of microbubble densities[40,45]. As shown in Fig. 2c, the maximum microbubble density that the PSF-CC method can determine is less than 6 microbubbles mm⁻². The identification drops exponentially to less than 5% when the microbubble density is larger than 75 microbubbles mm⁻² and less than 2% at a density of 200 microbubbles mm⁻². In contrast, the accuracy of the SC method drops linearly and slowly. The SC method maintains >80% of identification within 100 microbubbles mm⁻² and >60% when the microbubble density is less than 200 microbubbles mm⁻². When comparing these two methods, the SC method recovers up to 37 times as many microbubbles as the PSF-CC method. The localization precision curve (Fig. 2d) shows that, with the same high microbubble density, the SC method recovers microbubble locations with higher accuracy than the PSF-CC method. Nevertheless, it is obvious that both

methods lose localization accuracy when the microbubble density increases. Therefore, there is a trade-off between the temporal resolution and spatial resolution of ULM. In this study, we assume PSF has a Gaussian distribution and is spatially invariant to obtain analytic PSF expression. It is worth noting that PSF of actual conditions varies spatially, especially when the microbubbles become close to each other, thus, could lead to additional recovery error.

Our SC optimization uses a fast shrinkage-thresholding algorithm (FISTA)[46,47] to improve the computation efficiency (Methods). FISTA is a fast-proximal gradient method, which is an iterative method to solve optimization problems where some functions are non-differentiable, as in our case. It proposes a proximal operator to "mimic" the gradient of the non-differentiable functions and then operates as a gradient descent method. FISTA combines the proximal gradient step and the prediction step so that it can accelerate the convergence rate up to the first order. We compared the performance of our sparsity recovery method using FISTA with other methods, such as CVX[39] and L1-homotopy[40]. As shown in Supplementary Fig. S5, all sparsity recovery methods have a similar recovery performance and improve the recovery rate, especially for densely packed microbubbles, compared with the standard PSF-CC method. However, the computational time is the main disadvantage of prior sparsity recovery methods due to the requirement of patch-stitch image processing and the iteration algorithm[48]. The expensive operations of CVX and L1-homotopy limit them in computing large image sizes. As a result, they deal with small patches (a few tens of pixels) each time and stitch these patches to generate the final super-resolution image. Compared with CVX[39] and L1-homotopy[39,40], our SC approach is up to 57-fold faster in computational time than CVX and up to 10-fold faster than L1-homotopy at various microbubble densities (Supplementary Fig. S5a, the ultrasound image is $25 \times 25$ pixels and the final super-resolved image size is $200 \times 200$ pixels). It is worth mentioning that the computational time of these methods highly depends on the image size. Supplementary Table S2 shows that our SC approach could be even faster (137-fold) than CVX when dealing with large image sizes. The stitching may also decrease localization accuracy. Our FISTA approach uses lightweight operations instead and can rapidly solve sparsity recovery of the whole image (i.e., $200 \times 200$ pixels in Fig. 2a) while maintaining localization accuracy[46]. Supplementary Figure S2b shows that our SC method is more accurate than CVX and L1-homotopy by six-fold and five-fold, respectively.

### In vivo dual PA/fast ULM imaging of a mouse lymph node

We further investigated the feasibility of in vivo dual imaging of mouse lymph nodes. Lymphadenectomy, a micro-surgery to remove the lymph node invaded by cancer cells, can cause blood vessel injury[49]. It is critical to identify the blood vessels before and during the surgery. Further, the emerging data have indicated that the micro-vessel density around the lymph node positively associates with lymph node cancer metastasis in breast cancer[50]. Therefore, visualizing both lymph nodes and the surrounding micro-vessels becomes increasingly essential. Immunohistology of biopsy samples is still the gold standard for studying the status of lymph nodes and microvasculature. While PA imaging alone can visualize dye-labeled lymph nodes and large vessels around the lymph nodes, it cannot resolve the microvasculature[51,52]. Here, we demonstrate dual in vivo imaging of indocyanine green (ICG) dye-labeled popliteal lymph node (PLD) and the microvasculature around and within the lymph node in a mouse hindlimb.

We imaged a mouse hindlimb with the abovementioned dual-imaging system but PA mode off to evaluate the performance of SC-ULM. The lipid microbubbles with an average size of $1.036 \pm 0.005 \mu m$ are used for the ULM imaging (Supplementary Fig. S6 and Methods). After the DAQ, we processed the images using the SC method and regular PSF-CC method to generate ULM images (Methods). We also processed the same images using the power-Doppler method[53], a commonly used method to generate a diffraction-limited vascular image as a reference. Figure 3a and b shows the SC-ULM and PSF-CC ULM images generated at 1.5 sec of DAQ, as well as the reference power-Doppler image (Fig. 3c). As expected, the ULM images produce sharp and high-resolution microvascular features, and, as a comparison, the small blood vessels in the power-Doppler image are visually blurred. As we have observed from Fig. 2a and 2b, SC-ULM identifies more microbubbles, thus, is brighter than the PSF-CC ULM image.

To quantitatively compare the in vivo microbubble identification of SC and PSF-CC methods, we correlated the number of identified microbubbles with normalized pixel saturation in the images (Methods). We plotted the pixel saturation of SC-ULM and PSF-CC ULM images as a function of the DAQ duration (Fig. 3d); SC-ULM recovers the vessel features (saturated) much faster than PSF-CC ULM. SC-ULM almost recovers all saturated vessels (saturation = $0.98 \pm 0.008$) within 5 sec, while PSF-CC produces less than 50% of saturation simultaneously. The saturation curves are fitted through an asymptotic regression model with an asymptote at 1 (the solid curves in Fig. 3d). The PSF-CC ULM is much slower than SC-ULM; PSF-CC ULM cannot reach the same saturation level in an average of 12 sec as SC-ULM in 1.5 sec. We define a characteristic time of a saturation curve to quantitatively compare the imaging speed (the minimum required DAQ time)[36]. We first calculated the slope of the saturation curve at its origin. The characteristic time is the time required for the slope to reach 100% saturation. In PSF-CC ULM, the characteristic time can be calculated by extrapolating the slope to prevent a long DAQ time. The images of both methods reach similar saturations at their corresponding characteristic time; thus, the ratio of the characteristic times indicates the relative imaging speed of these methods. The characteristic times of SC-ULM and PSF-CC ULM are 1.5 and 32.5 sec, respectively, a 22-fold improvement for in vivo imaging speed using SC-ULM. Figure 3e shows the zoom-in micro-vessel images of a selected area from Fig. 3a (top row) and Fig. 3b (bottom row) at various DAQ times from 0.5 to 7 sec. It shows that the PSF-CC method reconstructs sparse and barely recognizable vascular structures until 7 sec of acquisition. The SC-ULM image has an almost complete reconstruction of the primary vascular structures within 0.5 sec.

We compared the resolution of SC-ULM and PSF-CC ULM images in Fig. 3e at 1.5 sec of DAQ time using Fourier ring correlation[54]. Using the half-bit threshold, the resolution of SC-ULM is 36.6 μm (Fig. 3f), while that of PSF-CC ULM is 35.8 μm (Fig. 3g). The slightly lower resolution mainly comes from the tissue noise. Unlike the simulation case, in vivo microbubble signal is much weaker than the tissue signal. The tissue noise signals could be falsely recognized as microbubble signals by both PSF-CC and SC. SC is more vulnerable than PSF-CC because it identifies more points in high-density microbubble cases, which would contribute to larger localization errors than the simulation and make the resolution of SC-ULM a little lower than that of PSF-CC ULM. We further analyze the resolution of each time point of micro-vessel images; our results show the resolution of the ULM images depends on the DAQ time. Although the SC-ULM has a slightly worse resolution than PSF-CC ULM starting at the beginning of the imaging due to the localization errors from high-density recovery (Fig. 2d), the difference in resolution between SC-ULM and PSF-CC ULM becomes insignificant after 1.5 sec (Fig. 3h).

SC-ULM accelerates the imaging speed, making it possible for a 3D ULM scanning with a linear transducer and co-register it with a 3D spectral PA image. Below we show 3D in vivo dual imaging of a mouse popliteal lymph node and the surrounding micro blood vessels in one scan. To visualize the lymph node, we first inject ICG dye solution (20 μL, 0.2 mg/ml) into the footpad of mice. ICG dyes drain and accumulate in the lymph node. The ICG accumulation is first validated using the IVIS system, where the lymph nodes show a higher fluorescence intensity (Fig. 4a). After 15 min of the ICG injection, we use the dual PA/SC-ULM system to image the blood vessels and lymph nodes.

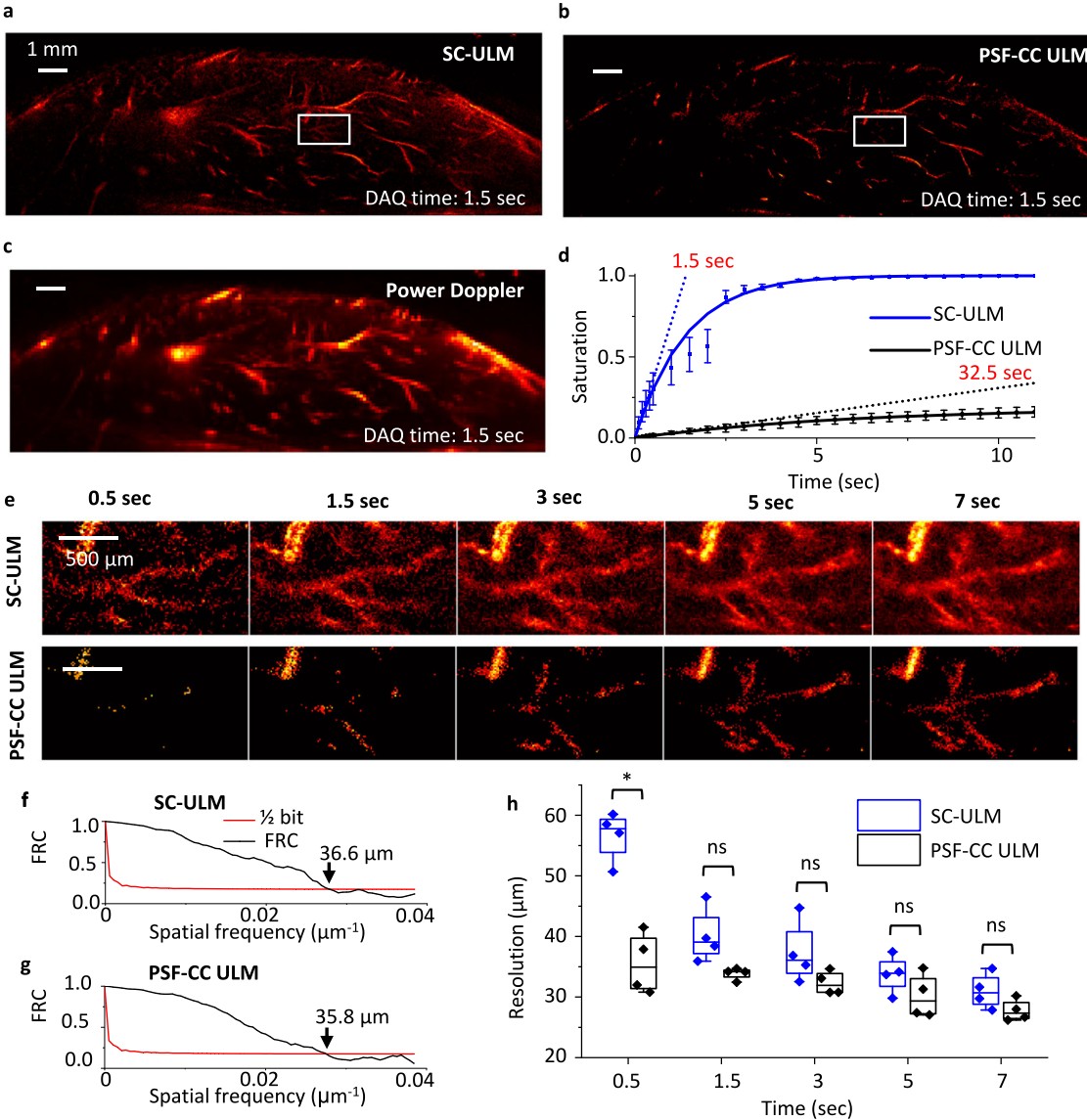

**Fig. 3 | The SC method enables in vivo fast super-resolution vascular imaging.** Super-resolution ultrasound vascular imaging of a mouse hindlimb using **a** SC-ULM, **b** PSF-CC ULM, and **c** power-Doppler methods. All the images are recorded from 1.5 sec of data acquisition (DAQ). **d** The image saturation as a function of DAQ times in SC-ULM (blue) and PSF-CC ULM images (black), respectively. The averaged saturation is calculated from 96125 pixels from three randomly selected regions of interest (ROI) in the vascular structures. The error bars are standard deviations ($N = 3$). The solid curves are fitted through an asymptotic regression model with an asymptote at 1. The characteristic times are 1.5 and 32.5 sec with SC-ULM and PSF-CC ULM, respectively. The latter is calculated through extrapolation because PSF-CC is slow and cannot reach the same saturation in 12 sec. **e** The zoom-in view of vascular structures using SC-ULM (top row) and PSF-CC ULM (bottom row) at various DAQ times from 0.5 to 7 sec, respectively. The area shown here is marked by the white rectangles in **a** and **b**. The result shows SC-ULM reveals the structure of major vessels at 0.5 sec and nearly complete structures in 1.5 sec, while the PSF-CC image remains sparse after 7 sec. **f, g** resolution measurements using the Fourier ring correlation (FRC) method of SC-ULM and PSF-CC ULM at 1.5 sec of DAQ in (**e**), respectively. **h** The resolution of both methods as a function of DAQ times. The averaged resolution is calculated from four random ROIs with the FRC method. The center line in each box is the median, and the bottom and top edges of the box indicate the 25th and 75th percentiles, respectively. The whiskers range of each box is within 1.5 interquartile. Scatter plots of the data used for the boxplot are overlaid on each boxplot. *$p = 0.0015$, ns denotes not significant.

A three-dimensional scan covers the whole hindlimb. Thanks to the acceleration of SC-ULM, less than 2 sec (1.5 sec for ULM and 0.3 sec for PA) are needed for each scanning position, and a total of 16 positions are captured with a step size of 100 μm. Figure 4b shows the 3D US image of the hindlimb, with no visible blood vessels or lymph nodes. PA imaging using multiple laser wavelengths (750, 790, and 850 nm) and a linear spectral unmixing method can separate the ICG signal from hemoglobin in tissue and reveal the position and shape of ICG-labeled lymph node and lymphatic vessels (Fig. 4c). Compared to PA imaging, 3D SC-ULM achieves a high spatial resolution of blood vessels near the lymphatic system (Fig. 4d). Since the PA and SC-ULM images are acquired from the same scan using the same imaging system, it is straightforward to co-register the two imaging modes to construct the combined 3D lymph node and microvasculature image (Fig. 4e). It is worth noting that when zoomed in to the lymph node region of the dual image, it shows that our imaging technique also identifies the lymph node microvasculature. (Fig. 4e, white box).

### In vivo dual PA/fast ULM imaging of a mouse kidney
We further demonstrate dual imaging of a deep-seated organ in mice. Renal oxygenation and hemodynamics are known to be associated with acute kidney injury[55]. Recent evidence suggests a pathogenic role

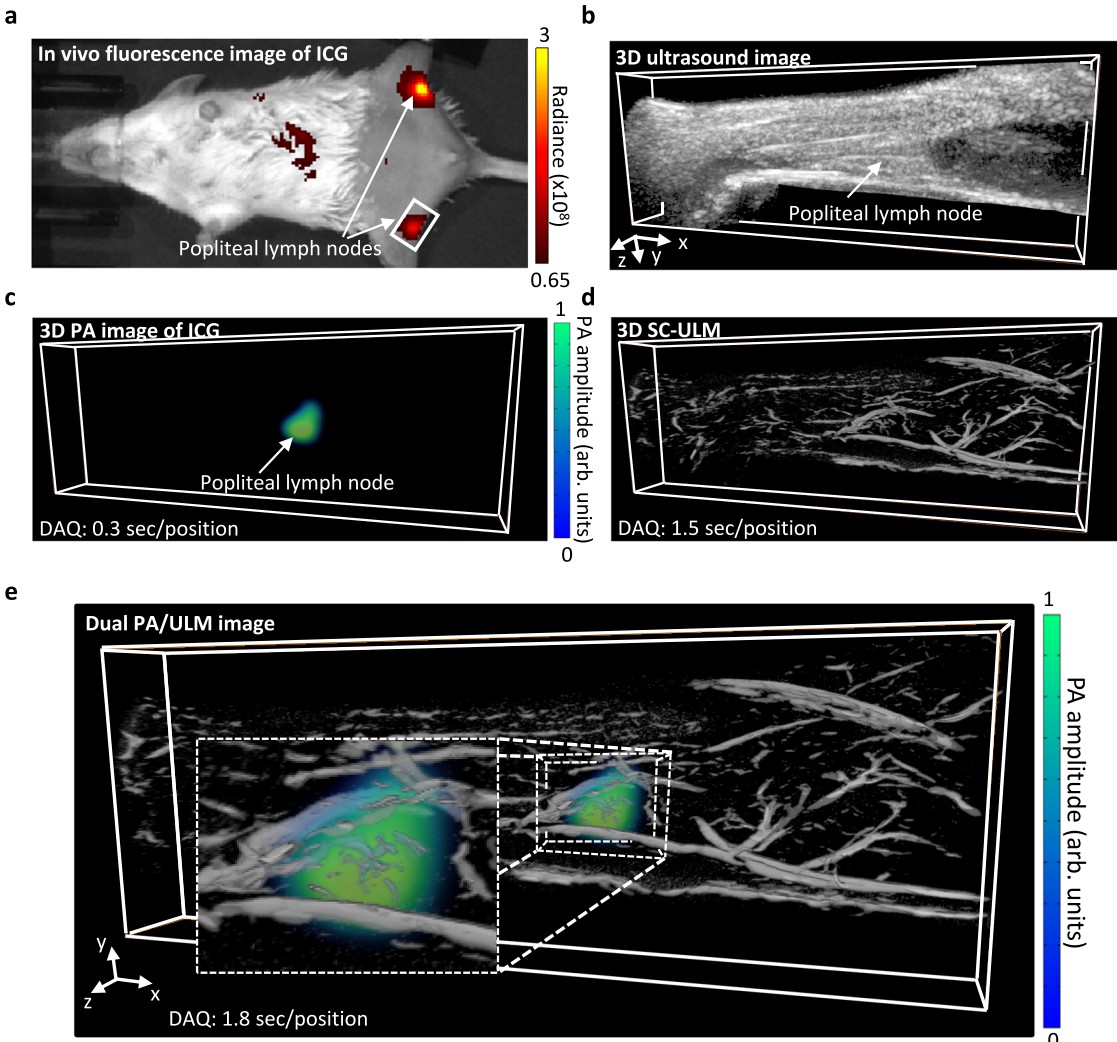

**Fig. 4 | In vivo dual PA/SC-ULM image of a mouse popliteal lymph node. a** Near-infrared fluorescence lymphatic imaging (IVIS) of indocyanine green (ICG) shows the locations of mouse popliteal lymph nodes (PLNs) where ICG accumulates. The ICG solution (20 μL, 0.2 mg ml⁻¹) was administered to both hind footpads and drained to PLNs within 15 min. The imaging area is 48 × 87 mm. **b** 3D ultrasound image of a mouse hind limp close to the popliteal fossa area. **c** 3D SC-ULM image shows the blood vessels within the hindlimb tissue near the PLN. **d** 3D PA imaging of ICG confirms the ICG accumulation in the PLN. Three laser wavelengths (750, 790, and 850 nm) are used for multi-wavelength PA imaging, and linear spectral unmixing is used for identifying ICG signal spectroscopically. **e** 3D dual PA/SC-ULM image. The PLN is labeled with ICG; the SC-ULM shows blood vessels surrounding/within the PLN. The 3D imaging volume in (**b**–**e**) is 22.9 mm (x) × 8.0 mm (y) × 1.6 mm (z).

of renal hypoxia in chronic kidney disease[56–58]. The crosstalk between kidney vasculature abnormalities to renal oxygenation is largely understudied. Kidney biopsy and BOLD-MRI are the few available choices to monitor renal tissue oxygenation[59,60]. However, a frequent biopsy may cause unnecessary complications; due to cost and the low accessibility of MRI, it is not an ideal imaging modality for longitudinal studies, especially in the preclinical setting. PA imaging was proven effective in detecting hemoglobin oxygenation; ULM is promising in revealing renal hemodynamics. Dual PA/ULM imaging can be a convenient tool for preclinical renal studies.

ULM imaging estimates the blood velocity by tracking the movements of the microbubbles, frame by frame. The physiological motions causing the error of microbubble tracking, thus, attribute to the error of localization and further the error of blood velocity estimation. The kidney is deeply embedded in the abdominal cavity, and tissue motions, such as respiration and cardiac pulsation, can induce motion and deformation of the kidney[61]. Figure 5a shows ULM images of a renal vascular tree reconstructed by SC and PSF-CC methods with 1 sec and 4 sec of DAQ, respectively. Compared to PSF-CC

reconstruction, as expected, the SC method identifies microbubbles fast and fills up the vascular structures, including microvasculature, within a second. At the same time, the PSF-CC image remains sparse, especially in the region of the small blood vessels where blood flow is slow, and a small number of microbubbles are present in the bloodstream (Fig. 5a and Supplementary Fig. S7a, b). When increasing the DAQ time to 4 sec to improve the imaging saturation, both SC and PSF-CC images, however, become blurred due to the physiological motions (Fig. 5a).

To identify and classify the motions, we calculated the frame-to-frame cross-correlation to a series of ultrasound kidney images (Supplementary Fig. S8b and Methods). The curve shows periodic respiratory cycles and the correlation damping out over multiple respiration cycles. We normalized the correlation of each respiratory cycle to show the respiratory motion details, including inspiration (breathing-in), expiration (breathing-out), and cardiac pulsations (Fig. 5b). To correct these motions, we assumed a rigid motion[61] to model the imaging targets' displacements, separating the physiological movements into lateral, axial, elevational, and rotational

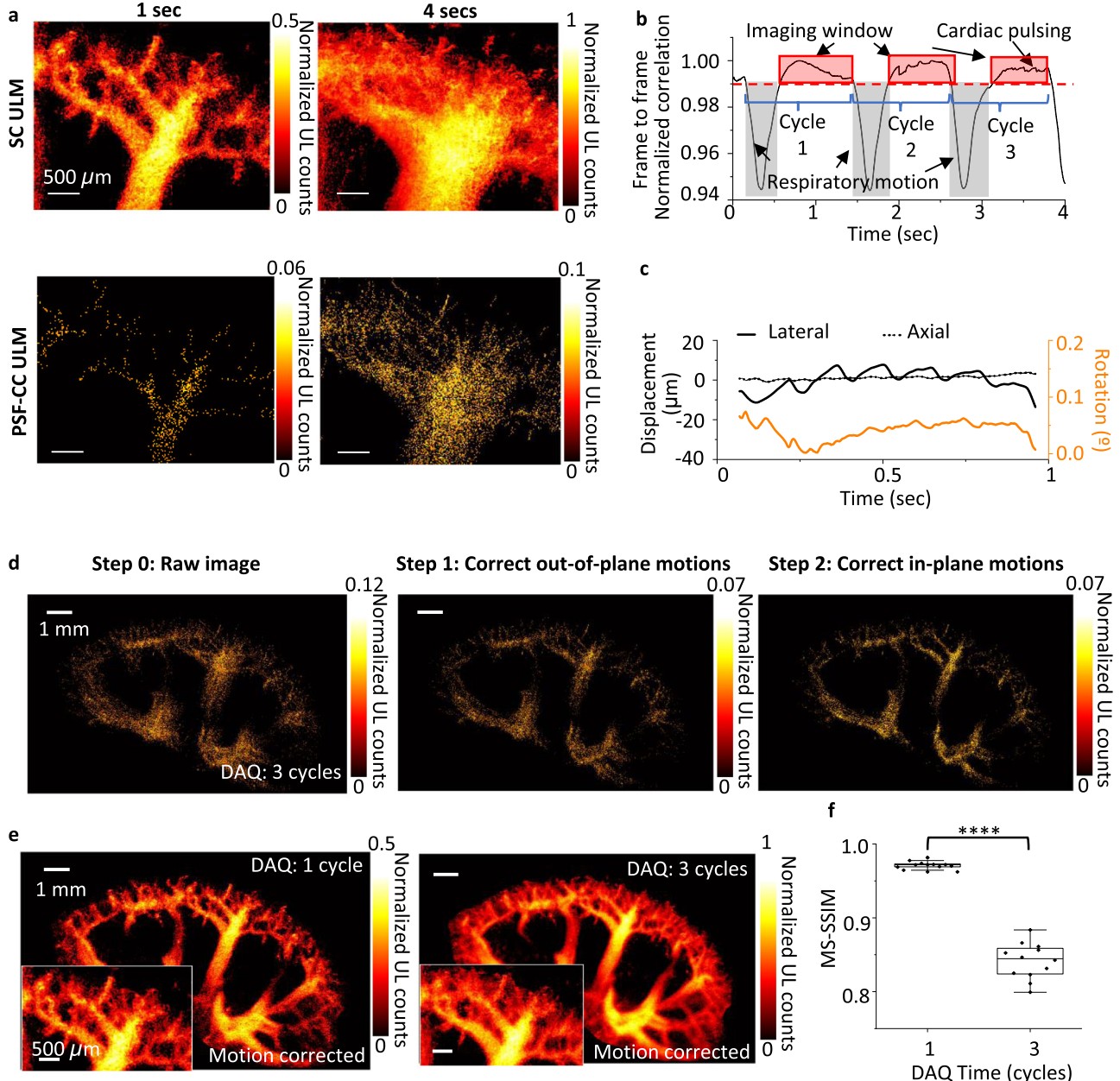

**Fig. 5 | Motion correction of in vivo mouse renal vascular imaging. a** The images show raw ULM images of a kidney arterial and venous renal tree acquired through 1 sec (first column) and 4 sec (second column) of DAQ time. The images are processed with SC (first row) and PSF-CC method (second row), respectively. SC-ULM image reconstructs renal vasculature within a respiratory cycle (~1 sec), and PSF-CC requires at least (3 respiratory, ~4 sec, to reconstruct a visible renal structure. The long DAQ time greatly degrades the image quality due to the motion. **b** Motion-induced frame-to-frame correlation changes during 4 sec of ultrasound image recording. The identified motions include respiratory and cardiac motions. **c** A representative in-plan motion within a 2D ultrasound image. The detectable in-plan motions contain lateral (solid black), axial (dotted black), and rotational (solid orange) components. **d** The comparison before and after applying out-of-plane (step 1) and in-plane (step 2) motion-correction algorithms to the PSF-CC ULM image. **e** The comparison of motion-corrected SC-ULM kidney images acquired with one breathing cycle of DAQ time and three breathing cycles of DAQ time. The insets show the zoom-in view of the renal vasculature trees. **f** The image similarity comparing SC-ULM images before and after the motion correction for one breathing cycle (~1 sec) and three breathing cycles (~4 sec). The center line in each box is the median, and the bottom and top edges of the box indicate the 25th and 75th percentiles, respectively. The whisker range is 5th to 95th percentiles. Scatter plots of the data used for the boxplot are overlaid on each boxplot. *p < 0.0001.

displacements (Supplementary Fig. S8c). Because a linear ultrasound transducer generates 2D images, within the sequential 2D images, lateral, axial, and in-plane rotational motions are tractable; elevational and out-of-plane rotational motions are non-tractable. The non-tractable motions contribute to localization errors. The first step of motion correction is to exclude the image frames containing non-tractable motions. From the normalized correlation curve, respiratory motions are prominent. These motions are usually large enough to

induce non-tractable out-of-plane motions, resulting in a significant correlation drop (Fig. 5b). We selected and discarded the frames with correlations under the threshold (0.95 used here) to exclude those affected by respiratory motions. We grouped the remaining frames into each recording cycle for images recorded longer than a respiration cycle.

We then corrected the in-plane motions through rigid-motion tracking. The representative lateral and axial displacements and

in-plane rotations of the first recording cycle are shown in Fig. 5c and Supplementary Fig. S8c. In a rigid motion model, we assumed the motion-affected (to-be corrected) frames are the replica of the reference frame with an in-plane translation and rotation. The corrected coordinates and rotational angles were calculated through a least squares method, optimized with a regular step gradient descent. Figure 5d shows the steps of ULM renal angiography motion correction by the PSF-CC method with 4 sec of DAQ time. We removed the respiratory motions from the image, then compensated for both intra-cycle and inter-cycle in-plane motions to produce a motion-corrected image. Compared to the raw image, the fine blood vessel structures are restored in the motion-corrected PSF-CC images (Fig. 5d). While the motion-correction algorithm dramatically improves the PSF-CC ULM images, it is worth noting that the errors from inter-cycle out-of-plane motions, the primary source to annihilate the correlation after each respiratory cycle, are accumulated over increasing cycles. In PSF-CC ULM, a long DAQ time is unavoidable; the residue localization errors from an out-of-plane motion are the side-effect of a long DAQ time. From the saturation curve of a super-resolution kidney image (Supplementary Fig. S4c), the characteristic time of PSF-CC ULM is more than 13 sec, which means that in practice, longer than 4 sec of DAQ time is required for PSF-CC to reconstruct a renal angiography and the out-of-plane errors will be more severe than 4 sec of the DAQ.

From the saturation curve (Supplementary Fig. S7c), the characteristic time of SC reconstruction in the kidney is 0.5 sec, 28 times faster than the PSF-CC reconstruction. Because of the vast size range of blood vessels in the kidney, many microbubbles fill out the large vessel structures first, resulting in more localization in the big vessels than in small vessels. This regional signal saturation falsely raises the averaged saturation of the whole image, although the small blood vessels are still under-constructed (Supplementary Fig. S7a–b). This nonuniform microbubble identification affects imaging resolution. We calculated the averaged resolution by FRC measurement from the SC-ULM kidney images at various DAQ times with motion corrections (Supplementary Fig. S9). In Supplementary Fig. S10, we summarized the resolution as a function of the DAQ times. Although the resolution of SC-ULM images is low at the beginning (at 0.5 sec) because of the insufficient small vessel reconstruction, it catches up after 1 sec of DAQ, within the duration of one respiratory cycle. The short recording time of the SC method offers a significant advantage in motion correction as it is less affected by the un-tractable out-of-plane motions that cause imaging degradation as in the conventional PSF-CC method.

We present two SC-ULM images of the kidney with 1 sec (~1 breath cycle) and 4 sec (~3 breath cycles) of DAQ, respectively (Fig. 5e and Supplementary Fig. S11), to demonstrate the advantage of the SC method in motion correction. We quantitatively compared the multi-scale-structural-similarity (MS-SSIM) between the images corrected with only the out-of-plane motions and those corrected with both out-of-plane and in-plane motions for both DAQs (Fig. 5f). The results show that with the fast recording, the 1 sec SC image is less affected by the inter-cycle motion. Hence, the in-plane correction shows minimal improvement on the 1 sec image (MS-SSIM = 0.97 ± 0.005), while the in-plane correction is needed for the 4 sec image (MS-SSIM = 0.84 ± 0.024). However, although the motion-correction algorithm removes the majority of motion artifacts (Supplementary Fig. S11b), in the 4 sec image, the result shows that the out-of-plane motions cause the untrackable shifting between cycles, resulting in the blurred image in fine vessel structures when compared with the 1 sec motion-corrected image. These results suggest that the SC method not only improves the imaging speed but also positively reduces the complexity of the motion correction. The fast SC-ULM enables the tracking of kidney blood vessels over breathing cycles, which is impossible for regular PSF-CC ULM imaging (Supplementary Fig. S12).

We produced images of renal vascular flow direction and speed using motion-corrected fast SC-ULM images by mapping the microbubbles' movement (Fig. 6a, b). It has been shown that renal blood flow velocity correlates to renal tissue oxygenation[62,63]. To showcase the capability of the interleaved PA/SC-ULM imaging, we conducted an in vivo oxygen-challenge experiment in mice to mimic various scenarios of tissue oxygenation and image the corresponding blood velocity change. During the imaging, the mouse inhaled 100, 3, and 20% of oxygen for 50 sec alternately. In Fig. 6c–f, we show that the interleaved scanning enables the frame-to-frame image co-registration of the renal blood speed distribution and the renal hemoglobin oxygenation distribution in the three levels of oxygen inhalation. In the highlighted regions of Fig. 6b, the zoom-in blood velocity maps show the change of blood flow speed in highlighted blood vessels when the oxygen inhalation level changes (Fig. 6g). We quantified the blood flow speed and hemoglobin oxygenation in the highlighted regions and plotted their differences as a function of the recording time (Fig. 6e, f). The result indicates a positive correlation between blood flow speed and hemoglobin oxygenation. This suggests our interleaved PA/SC-ULM is a valid imaging tool to non-invasively monitor the renal oxygenation and hemodynamics in mouse kidneys simultaneously.

## Discussion

This study shows a dual-contrast PA/SC-ULM imaging with an interleaved acquisition scheme. The dual-imaging system is developed using a widely used linear array ultrasound imaging system. The resulting PA/SC-ULM imaging is a non-ionizing, portable, and cost-effective imaging technique capable of super-resolution vascular and tissue physiological imaging in vivo. To overcome the inherently slow ULM, we significantly improve the imaging speed of the super-resolution US imaging using an SC optimization method. Our in vivo results show that the frame rate of SC-localization is ~28 times faster than that of conventional PSF-ULM. The ULM acceleration reduces the DAQ time to less than 2 sec, releasing the constraint of complicated motion correction and further enabling the interleaved PA/ULM 3D scanning.

Conventional dual-imaging conducts each 3D scan sequentially. Because of the un-trackable elevational motions in an ultrasound linear array system, the motion errors accumulate between the frames recorded from different breathing cycles. The severe un-trackable inter-breathing cycle artifacts and mechanical scanning errors can cause misalignment between 3D image sets, complicating the image registration process. With an interleaved recording and the fast SC-ULM, we ensure each PA and ULM frame is recorded within one breathing cycle, which greatly minimizes the effect of untrackable motions. Our results show that interleaved imaging significantly reduces the burden of the spatial registration of the 3D PA and ULM imaging sets. The interleaved recording is also crucial to 2D time-lapse imaging. The alternate recording of PA and ULM images over time minimizes the un-trackable motions between two images and improves the temporal and spatial registration of two time-lapse imaging sets.

While the dual PA/SC-ULM imaging shows an unprecedented capability, it can be further improved. For example, one can upgrade the hardware and software further to enhance the speed of the dual-modality technique. The laser repetition rate limits our current setup's PA imaging frame rate. Using a laser with a higher repetition rate (up to 100–1000 Hz)[8] or combining multiple high repetition rate but low pulse energy lasers can improve the speed of PA imaging. In software, the speed of SC-ULM pre-processing (i.e., clutter filtering) and post-processing (i.e., sparsity recovery) have room to improve. Recently proposed real-time clutter filtering using random singular value decomposition and random spatial downsampling can alleviate the problem[64]. In addition, GPU-accelerated FISTA can be used for further

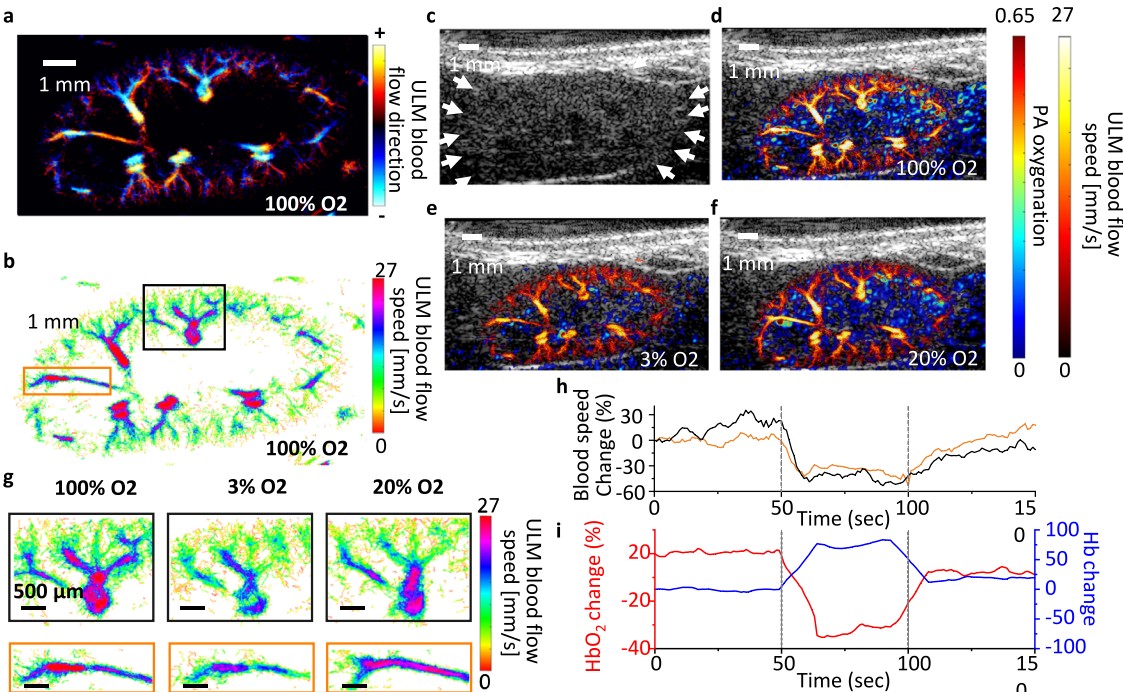

**Fig. 6 | In vivo renal oxygenation and vascular image of a mouse kidney in an oxygen-challenging test. a** Directional blood flow map and **b** blood speed map of a mouse kidney when the mouse inhales 100% oxygen. **c** An ultrafast plane-wave ultrasound image of a mouse kidney. **d**–**f** The dual-contrast images show co-registered PA renal oxygenation image and ULM blood speed image, recorded in **d** 100%, **e** 3%, and **f** 20% of oxygen inhalations. **g** The zoom-in ULM blood flow speed maps represent the highlighted regions in (**b**) under various levels of oxygen inhalations. **h** Quantitative analysis of the blood speed change as a function of time over 150 sec of recording. **i** Quantitative analysis of the hemoglobin oxygenation changes as a function of time over 150 sec of recording. The red (and blue) curve represents the percent change of oxygenated (and deoxygenated) hemoglobin. The ultrasound image is recorded with a frame rate of 500 Hz and a central frequency of 15 MHz.

acceleration[65]. We predict that with these strategies, it is possible to accelerate dual-modality imaging into real-time implementation.

In conclusion, we demonstrate a dual PA/super-resolution US imaging technique and its applications in scenarios that are challenging to image using other modalities. We show that this imaging technique can generate dual-contrast images to reveal a dye-labeled lymph node as well as micron size blood vessels within and near the lymph node. We also demonstrate that the imaging maps the renal oxygenation and the blood flow of the microvasculature in a mouse kidney over 2 min. We expect this dual-imaging modality will promote a broad interest in cancer, neuroscience, nephrology, and immunology imaging applications.

## Methods

### Construction of the imaging system

The dual-modal PA/US system consists of a Verasonics ultrasound imaging research system (Vantage 256, Verasonics Inc., Kirkland, WA, USA), a 15 MHz linear array transducer (MS200, Visualsonics Inc., Toronto, ON, Canada), and a wavelength tunable (690–950 nm) OPO laser source with 7-ns-pulse and 10 Hz pulse repletion rate (Phocus Essential, Opotek, Inc., Carlsbad, CA, USA). The laser fluence used in the study is 15 mJ cm$^{-2}$. A customized bifurcated fiber bundle is used to deliver light from the laser source. The fiber bundle is integrated into the transducer using a 3D-printed adaptor for the side illumination. When performing imaging, the transducer is connected to Verasonics hardware for data acquisition of ultrasound and photoacoustic signals. We use a function generator to synchronize the data acquisition and the laser firing.

### Ultrasound simulation

Verasonics Ultrasound Simulator[66] was used to generate a raw data cineloop. In the simulation mode, a 15 MHz linear array transducer was applied to transmit plane waves with seven angles (−6° to 6°) at a pulse repetition frequency of 500 Hz. We used different numbers of points randomly distributed in a fixed region with amplitudes following Gaussian distribution to represent microbubble signals. For each microbubble density, the simulation was repeated 20 times. The received RF data was then reconstructed to a B-mode image using the delay-and-sum algorithm. A total of 1000 frames were recorded. The image data were processed by both sparsity-constrained optimization and PSF cross-correlation methods to recover the microbubble locations.

### Sparsity-constrained optimization and PSF cross-correlation

The high-pass spatiotemporal filtering[53] was applied to in-phase/quadrature (IQ) data to extract the flowing microbubble signals after data acquisition. After filtering, thresholding was applied to further remove noises. Since the bubbles are much smaller than the wavelength and are individually separated in space and time, their pattern can be considered the point spread function (PSF) of the ultrasound system. We consider human and animal soft tissues homogeneous for acoustic properties at the first-order approximation. The recorded PSF can be approximated to the theoretical PSF calculated from the theory of acoustic diffraction. A recorded PSF in our studies was estimated using Gaussian fitting of one isolated microbubble pattern[28,32].

We treat each decluttered ultrasound image frame as the down-sampled and blurred image of the super-resolved ultrasound image. The diffraction-limited ultrasound image is the convolution between the PSF of the ultrasound system and microbubble distributions:

$$b = DHx + w, \qquad (1)$$

where **b** is a vectorized ultrasound image for ultrasound localization microscopy, **D** is a downsampling operator, and **H** is a blurring

operator, which can be represented as a block circulant with circulant block (BCCB) matrix related to the PSF of the ultrasound system, **x** is a vector related to microbubble distributions (i.e., the image of microbubble positions to be recovered), and **w** represents the noise.

For sparsity-constrained optimization, considering the sparsity of microbubble distribution over space, we can recover the microbubble distribution via sparsity regularization using ℓ1-minimization[48]. In addition, because negative microbubble intensity is non-physical, we restrict that the values of x cannot be negative. Therefore, we have the following ℓ1-minimization:

$$\hat{\mathbf{x}} = \arg \min_{x \geq 0} ||\mathbf{DHx} - \mathbf{b}||^2 + \lambda ||\mathbf{x}||_1 = \arg \min_{x \geq 0} ||\mathbf{Ax} - \mathbf{b}||^2 + \lambda ||\mathbf{x}||_1 \quad (2)$$

where $\lambda$ (>0) is the regularization parameter controlling the influence of the penalty on the estimate. The downsampling factor is set to 8 in this paper.

Here, we used a fast shrinkage-thresholding algorithm (FISTA)[46,47], a fast-proximal gradient method, to solve the above convex optimization problem. The proximal gradient method is an iterative method trying to solve the optimization problem where some functions are non-differentiable (for example, $\lambda ||\mathbf{x}||_1$ in our case). It proposes a proximal operator to "mimic" the gradient of the non-differentiable functions and then operates as a gradient descent method. FISTA combines the proximal gradient step and the prediction step so that it can accelerate the convergence rate up to the first order. The method is named FISTA because the proximal operator of the ℓ1 norm function is the shrinkage operator $T_\alpha(\mathbf{x})_i = (x_i - \alpha)_+ \text{sgn}(x_i)$. In our case, the shrinkage operator is modified to only consider the non-negative x.

**Algorithm 1:** FISTA

Input: One vectorized frame **b**, regularization $\lambda$(>0)
Output: MB distribution **x** for one frame
Step 0. Take $\mathbf{p}_1 = \mathbf{x}_0 = \mathbf{0} \in \mathbb{R}^N, t_1 = 1, \tau = 1/||\mathbf{A}^T\mathbf{A}||$
Step k.

$$\mathbf{x}_k = T_{\lambda\tau}\left(\mathbf{p}_k - 2\tau\mathbf{A}^T(\mathbf{A}\mathbf{p}_k - \mathbf{b})\right), \text{where } T_\alpha(\mathbf{x})_i = (x_i - \alpha)_+,$$

$$t_{k+1} = \frac{1 + \sqrt{1 + 4t_k^2}}{2},$$

$$\mathbf{p}_{k+1} = \mathbf{x}_k + \left(\frac{t_k - 1}{t_{k+1}}\right)(\mathbf{x}_k - \mathbf{x}_{k-1}).$$

In Algorithm 1, it involves the application of $\mathbf{A}^T\mathbf{A}$ and $\mathbf{A}^T$ on vectors in step k. In a real experiment, the matrix $\mathbf{A} = \mathbf{DH}$ can have a very large size (~10,000 × 640,000), and it could be inconvenient to store and do **A** related calculations. For other sparsity-constrained approaches, like CVX and L1-homotopy, instead of dealing with the whole image, it only picked up a small region (7 × 7 pixels) once and patched recovered results to generate the final super-resolved image. Therefore, the matrix **A** is quite small (~49 × 3136). In this paper, thanks to the properties of **H** as BCCB matrix, we did not store the whole big matrix **A** in memory, but we used FFT operation to apply $\mathbf{A}^T$ or $\mathbf{A}^T\mathbf{A}$ on vectors directly with efficient computation.

For the comparison, the CVX optimization package[67,68] and L1-homotopy package[69] in MATLAB were used to perform other sparsity-constrained approaches (i.e., CVX and L1-homotopy). In addition, we used PSF-CC to generate regular ULM[32]. Briefly, after clutter filtering and denoising, a normalized 2D cross-correlation is applied with the estimated PSF from Gaussian fitting to identify microbubbles in each frame. The threshold of the cross-correlation coefficient is set to 0.6.

After obtaining all microbubble locations over frames using either sparsity-constraint approaches or standard localization methods, a motion-correction algorithm is used to correct microbubble positions[61]. Then moving microbubbles are tracked with adjacent frames to discard uncorrected localized positions using a particle tracking algorithm (simpletracker.m available on Mathworks)[28]. The tracking algorithm is based on the Hungarian method[70]. The algorithm calculates all distances between all microbubbles in the current frame and all microbubbles in the next frame. Then the algorithm minimizes the total distance and finds the optimal pairs of microbubbles in adjacent frames. Applying the algorithm to all frames can give us the collections of a series of microbubble tracks. Because of the fast blood flow speed, microbubbles cannot always remain in the field of view. Instead, some bubbles flow out of the field of view, and some could collapse during imaging. New microbubbles flow into the field of view to produce new traces. What we have traced is each microbubble during certain frames, and we accumulated all traces to construct the super-resolution frames. The super-resolved velocity map can also be accessed by calculating the velocities of each microbubbles using the distance of trajectory and the time interval between frames.

### Imaging agent synthesis
The microbubble synthesis was adapted from[61]. Briefly, DSPC and PEG with a mass ratio of 1.2 mg: 0.58 mg were mixed in 1 mL chloroform in a small vial. The solution was transferred to a 100 mL round bottom flask, and then all the chloroform was evaporated for ~30 min, and the lipid cake was formed using a rotary evaporator (RE-502). After the evaporation procedure was completed, 1 mL of PBS (phosphate buffer saline) with 10% glycerol solution was added to the vial and sonicated until all the lipid cake dissolved into the solution. Then the solution was transferred to a sealed sterile vial. To activate the microbubble solution, approximately 12 mL of perfluorobutane gas (FluoroMed, LP) was transferred to the vial using a plastic syringe, and then the vial was placed in a vial mix (Lantheus) for 45 sec.

For photoacoustic contrast agents, ICG was purchased from ATT.

### Imaging agent characterization
The size and concentration of the microbubbles were measured by a particle sizing system instrument (AccuSizer FX Particle Sizer, Entegris, Inc., MA, USA). We measured size and concentration using 30 mL of microbubble/PBS solution, which was diluted 500 times with PBS from the as-prepared microbubble solution. Each size and concentration characterization were repeated three times to minimize measurement uncertainty. The mean size of the microbubbles was 1.03 μm (Supplementary Fig. S2). A microplate reader (BioTek, Synergy) was used to measure the ultraviolet to visible extinction optical extinction spectra of the ICG solution.

### Phantom imaging
To prepare the tissue-mimicking phantom, a mold with 10 mm diameter rods was 3D printed. A 5% w/v agar phantom was made with cylinder inclusion first. Then 200 μL 1 × 10^6 bubbles/mL microbubble solutions with the same concentration were mixed with the same amount of polyacrylamide gel (1:1) and filled in the inclusion separately. The ultrasound image of the phantom was imaged with a 15 MHz ultrasound transducer.

### In vivo imaging of photoacoustic/ultrasound localization imaging
All animal experiments were approved by the Institutional Animal Care and Use Committee (IACUC) at the University of Illinois Urbana-Champaign. A total of 6 female BALB/cJ mice (Jackson Laboratory) at age 8–10 weeks were used.

The left kidney of the mouse was imaged using a 15-MHz ultrasound/photoacoustic transducer. The mouse was placed on the heat pad and anesthetized using 2% isoflurane at 1 L/min of oxygen flow. 100 μL of imaging agents/PBS solution (1 × 10^9 particles/mL) were

injected into the mouse through the tail vein. The dual-modal imaging was acquired as follows: For each imaging sequence, ultrafast ultrasound imaging was applied first through seven-angle (−6° to 6°) plane-wave transmission to collect 0.8 sec of data with the pulse repetition frequency of 500 Hz. Then, multi-wavelength photoacoustic imaging was performed to collect 0.2 sec of data with a pulse repetition frequency of 10 Hz. The laser wavelength was set to 750 and 850 nm for the first and second acquisitions, respectively. The raw RF data was then reconstructed offline using default Verasonics imaging reconstruction to get ultrasound and photoacoustic images (the image size is 50 × 135 pixels, and the pixel size is 100 µm). Photoacoustic images were processed using linear spectral unmixing to extract hemoglobin and deoxyhemoglobin signals. The oxygen saturation was calculated by the ratio of the hemoglobin signal over the summation of the hemoglobin and deoxyhemoglobin signals. Ultrasound images were further processed to get super-resolution ultrasound images.

The oxygen-challenge protocol of kidney imaging was as follows. During the imaging, 100% oxygen was used for 50 sec, then a mixture of 3% oxygen and 97% nitrogen was used for 50 sec, and finally, the mixture was changed to 20% oxygen and 80% nitrogen.

For dual-modal photoacoustic and ULM of hindlimb imaging, 20 µL ICG solution (0.2 mg/mL) was injected through the footpad of the mouse. After 15 min of the injection, the mouse was placed on the heat pad and anesthetized using 2% isoflurane at 1 L/min of oxygen flow. Then 200 µL of imaging agents/PBS solution (1×10⁹ particles/mL) were injected into the mouse through the tail vein. The 15 MHz transducer was scanned laterally to collect 3D dual-modal images. Totally 16 positions were collected with a step size of 0.1 mm. For each position, the dual-modal imaging was acquired as follows: ultrafast ultrasound imaging was applied first through seven-angle (−6° to 6°) plane-wave transmission to collect 1.5 sec of data with a pulse repetition frequency of 500 Hz. Then, photoacoustic imaging was performed to collect 0.3 sec of data with a pulse repetition frequency of 10 Hz, and laser wavelength was set to 750, 790, and 850 nm, respectively. The post-processing of raw data is the same as in kidney imaging (The ultrasound and photoacoustic images have a size of 60 × 175 pixels, and the pixel size is 100 µm).

To validate that ICG was delivered into the lymph node, we used an IVIS spectrum imaging system (PerkinElmer) to check the ICG biodistribution. An excitation filter centered at 745 nm and an emission filter centered at 800 nm with 2 sec of exposure time were used. We quantitatively analyzed the images using the Living Image 4.5 software.

## Motion-compensation

The B-mode stack was analyzed to estimate motion. As shown in ref. 61, there are two main sources of motion in kidney imaging: respiration and cardiac pulsation. Selecting a reference B-mode image and applying frame-to-frame normalized cross-correlation (corr2 function in MATLAB) to a series of ultrasound kidney images, one can identify frames with respiration motion, which has a relatively low correlation index and cardiac pulsation, which happens in each inter-breath cycle. Under the assumption that the respiration motion is large enough to produce out-of-plane motion, we can discard these frames to remove the respiration motion (out-of-plane motion) by setting the threshold to 0.99. The remaining frames can be characterized as different groups based on respiration cycles. Inter- and intra-cycle motion can be compensated using a rigid motion model. The rest frames are rotated and translated as a replica of the reference frame. We applied a frame-to-frame rigid transformation to the rest frames and calculated the translation and rotation information using gradient descent optimization (imregtform function with regular step gradient descent optimizer in MATLAB). Then we can use the information to correct microbubble localizations.

For hindlimb imaging, only frame-to-frame normalized cross-correlation was used to remove the respiration motion.

## Photoacoustic spectrum unmixing

The measured photoacoustic signal after optical fluence normalization is a linear combination of the spectral signatures of different chromophores (i.e., hemoglobin, deoxyhemoglobin, and exogenous contrast agents) with the relative concentration of each chromophores[12]:

$$P'(\lambda_i, x, y) = \varepsilon_{HbR}(\lambda_i)C_{HbR}(x,y) + \varepsilon_{HbO_2}(\lambda_i)C_{HbO_2}(x,y) + \varepsilon_{exo}(\lambda_i)C_{exo}(x,y), \quad (3)$$

where $P'(\lambda_i, x, y)$ is the photoacoustic signal recorded at a wavelength $\lambda_i$ after optical fluence normalization, $\varepsilon_{HbR}(\lambda_i)$, $\varepsilon_{HbO_2}(\lambda_i)$ and $\varepsilon_{exo}(\lambda_i)$ are the known molar extinction coefficients. $C_{HbR}(x,y)$, $C_{HbO_2}(x,y)$ and $C_{exo}(x,y)$ are the molar concentrations of different chromophores. By measuring PA signal at different wavelengths and using the spectrum of the known molar extinction coefficients, we can use linear least square methods to extract the molar concentration of different chromophores:

$$\begin{bmatrix} C_{HbR}(x,y) \\ C_{HbO_2}(x,y) \\ C_{exo}(x,y) \end{bmatrix} = (\varepsilon^T \varepsilon)^{-1} \varepsilon^T P', \quad (4)$$

where $P' = \begin{bmatrix} P'(\lambda_1, x, y) \\ P'(\lambda_2, x, y) \\ \vdots \\ P'(\lambda_N, x, y) \end{bmatrix}$ and $\varepsilon = \begin{bmatrix} \varepsilon_{HbR}(\lambda_1) & \varepsilon_{HbO_2}(\lambda_1) & \varepsilon_{exo}(\lambda_1) \\ \varepsilon_{HbR}(\lambda_2) & \varepsilon_{HbO_2}(\lambda_2) & \varepsilon_{exo}(\lambda_2) \\ \vdots & \vdots & \vdots \\ \varepsilon_{HbR}(\lambda_N) & \varepsilon_{HbO_2}(\lambda_N) & \varepsilon_{exo}(\lambda_N) \end{bmatrix}.$

In the experiment, the pulse energy was measured by the power meter, and the optical fluence was calculated by the pulse energy over the illuminated surface area. For the hindlimb experiment, the exogenous contrast agent was ICG. For the kidney experiment, there was no exogenous contrast agent used, therefore $C_{exo}(x,y)$ was set to 0.

## Data processing, analyses, statistics, and reproducibility

The data collection and signal processing were performed on a Dell workstation (Dell Precision 5820) with an Intel® Xeon® W-2155 Processor and 96 GB RAM.

We used MATLAB to process the images acquired with the Verasonics imaging system. The ultrasound images were shown in dB scale, and the photoacoustic images were in linear scale. 3D volumetric images in Fig. 4 (main text) were 3D rendered using Amira 2021. Data plot, average, and standard deviation were computed in Origin 2020. Each imaging experiment shown in the main text was repeated at least three times independently with similar results.

In the simulation, the localization error was estimated using the method from ref. 45. The recovered microbubble positions were matched with the closest true positions, and the offsets in either lateral or axial direction were calculated. For a fixed microbubble density, the histogram of offsets (See Fig. S3) was fitted with a Gaussian function and FWHM value as the localization error at that microbubble density.

The saturation curve was measured using the methods in ref. 36. Five images of interest (ROI) were selected manually, and then the saturation at a given time was defined as the ratio of explored pixels at that time and the overall pixels in ROI. We used the characteristic time, which is calculated as the slope of the saturation curve at its origin, to quantitatively evaluate the saturation behavior.

The image resolution was measured using Fourier Ring Correlation[54]. Briefly, the list of tracking was randomly separated into two parts, and two super-resolution images were generated based on two tracking data. Then the normalized correlation of two spectra of these images was calculated and a fixed threshold (1/2 bit) was used to determine the resolution. In addition, to evaluate the difference in resolution of ULM and SC-ULM, we calculated the two-tailed p-value

using an unpaired student *t*-test to determine the significance. We considered our data to be statistically significant with $p < 0.05$.

The curve of blood velocity change was plotted as follows. Two regions of the kidney were selected, as shown in Fig. 6g and the absolute velocity in these two regions was calculated by averaging each speed value in each region, respectively. The absolute velocity at 0 sec (the start of the oxygen challenge) was used as the reference to calculate the change of the velocity at these two regions.

The curve of Hb/HbO$_2$ change was plotted by calculating the Hb/HbO$_2$ change of the kidney during the oxygen challenge. Briefly, the region of the kidney was defined by manually segmenting the kidney boundary from the B-mode image (Fig. 6c). Hb/HbO$_2$ of the kidney was calculated by averaging the value of Hb/HbO$_2$ inside the kidney boundary. The mean value of Hb/HbO$_2$ of the kidney at 0 sec (the start of the oxygen challenge) was used as the reference to calculate the change of Hb/HbO$_2$.

## Data availability
The datasets generated during and/or analyzed during the current study are available from the corresponding author upon reasonable request.

## Code availability
The codes generated during and/or analyzed during the current study are available at https://github.com/illinois-mirl/Hybrid-Photoacoustic-and-Fast-Super-Resolution-Ultrasound-Imaging.

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

## Acknowledgements

We would like to thank Dr. King Li for his generous support in equipment and B.-Z. Lin and M. Strzebonski for their assistance in experiments. This work was partly supported by grants from Google Faculty Research Award, Jump ARCHES endowment through the Health Care Engineering Systems Center, Dynamic Research Enterprise for Multidisciplinary Engineering Sciences (DREMES) at Zhejiang University and the University of Illinois Urbana-Champaign, and NIGMS R21GM139022.

## Author contributions

Y.S.C. conceived the original idea. Y.S.C. and S.S.Z. designed the experiments. S.S.Z. performed the simulation and in vivo imaging. J.H. and R.J. contributed to microbubble synthesis and imaging agent characterization. Y.S.C., S.S.Z., Y.Z., and C.H.W. contributed to the discussion of the data and experimental results. S.S.Z., Y.S.C., and Y.Z. drafted the manuscript and all authors contributed to the writing of the manuscript. YSC supervised the entire study.

## Competing interests

The authors declare no competing interests.
