## [Peer review file · Nature Communications]

REVIEWER COMMENTS

Reviewer #1 (Remarks to the Author):

Zhou and Chen report sparsity constrained (SC) optimization to accelerate the frame rate of ULM. There is a 30- to 40-fold improvement, which warrants publication in the journal. The images are quite striking and all needed controls are present. I recommend publication after these minor issues are addressed.

1. The text should be reviewed by a native English speaker.
2. The authors mention a PA acquisition time of <0.5 seconds; this per wavelength?
3. Please include details on laser pulse width and fluence.
4. The US acquisition time is 1 second and the bubbles are in the vasculature. Isn't there significant movement of the bubbles over this one second? Blood flow is quite fast.
5. The authors mention that the algorithm identifies 92 of 100 bubbles. What happens to the other 8? They are lost to overlap or are out of depth or?
6. Figure 2a: One would think having a lower bubble density would lead to better counts. Lots of overlap here. Why does the PSF function behave so poorly?
7. Figure 2c is important and illustrates the value of the work.
8. Some more explicit details on total imaging time are needed. The contributions of both signal acquisition and signal processing are needed.
9. One of the key attributes of acoustic imaging is that it offers video frame rates relative to techniques like PET or CT. Some more comments on the slow frame rates and ideas to improve are needed in the discussion.
10. Might the value of interleave imaging be integrated for even further refinements? Some references and comments are needed.
 - a. <https://www.nature.com/articles/s41467-021-20947-5>
 - b. <https://opg.optica.org/abstract.cfm?uri=ol-47-14-3503>

Reviewer #2 (Remarks to the Author):

Summary:

This work proposes a dual PA/ULM technique for vascular imaging and physiological imaging computed at 2 second intervals. The SRUS method of sparsity is used to accelerate saturation of the vessel images. The methods are demonstrated in vivo. The novelty of the paper seems to be the use of PA interlaced with ULM. Only one image is shown as PA/ULM, that of the dye labeled mouse lymph node “with nearby microvasculature”. The kidney work is not displayed as PA/ULM, perhaps because the information does not fuse. The case for a dual mode of interlaced operation of PA and ULM, versus a sequential one, is not clear.

The authors state that the sparsity based method for ULM is novel, however, the case is not well founded. The conclusion of the paper, mainly resting on “development of a sparsity-constrained” ULM method with computational speedup does not stand.

Overall, the paper is well organized and well written.

There were some grammatical errors (not noted), but it did not impede the review.

Major Comments:

The authors claim:

However, SUSHI requires a high frame rate to resolve fine vessel structures, and the resolution is lower than localization based super-resolution methods.

And

However, the proposed compressed sensing methods for localization used patch-stitch image processing, which may lose computational speed and accuracy.

The reference cited (38) shows an algorithm (Algorithm 1) which is nearly identical to the one the authors provide in the supplement. Would the authors please describe where there are meaningful differences.

While this reference (38) used a high frame rate, and benefited from that, it is not stated as required for the method. Many high quality ULM images have been made using frame rates of hundreds of Hz.

There appears to be comparable accuracy of the SUSHI with FISTA method (38) with super localization – shown in Figure 3 of that work.

The simulation methods showing the SC method recovers more MB is not novel.

Detailed Comments

Manuscript:

p. 15 The section on performance in vivo being different is weak. The biological mechanisms leading to great implications on performance in realistic environments appear to be poorly understood. Phase changing nanodroplets are not the solution to “unbalanced vessel sizes”. The droplets may extravasate and tracing the vasculature may be more problematic.

p. 15 Nanodroplets are not a solution to MB dosage concerns, they are not yet FDA approved for instance.

Various passages use present tense when past is more natural.

References (many) are incomplete – lacking journal name.

Supplementary information:

Please provide a brief description of the algorithms for motion correction and tracking with references.

Various passages use present tense when past is more natural.

Please provide references for linear spectral unmixing.

Figure S2 – lower row, right image – missing the red curve or not visible, it may lie near y-axis.

References 6, 7, 9, 11 are incomplete – lacking journal name.

Response to Reviewers' Comments on manuscript **NCOMMS-22-36128** entitled "**Hybrid Photoacoustic and Fast Super Resolution Ultrasound Imaging**" by Shensheng Zhao, Jonathan Hartanto, Ritin Joseph, Cheng-Hsun Wu, Yang Zhao, Yun-Sheng Chen*

Reviewer #1:

General Comment:

Zhou and Chen report sparsity constrained (SC) optimization to accelerate the frame rate of ULM. There is a 30- to 40-fold improvement, which warrants publication in the journal. The images are quite striking, and all needed controls are present. I recommend publication after these minor issues are addressed.

Authors: We are thrilled about the very positive comments from the reviewer!

1. The text should be reviewed by a native English speaker.

Authors: We thank the reviewer's suggestion. We have sent the manuscript to our colleagues at the language center for a proof-read.

2. The authors mention a PA acquisition time of <0.5 seconds; this per wavelength?

Authors: The acquisition time of <0.5 seconds is for five wavelengths. In our system, one PA acquisition with one wavelength only takes 0.1 seconds, which is limited by the repetition rate (10 Hz) and wavelength switching of our laser system. This technical limitation of our current setup is not the theoretical limit of our spectroscopic PA imaging.

3. Please include details on laser pulse width and fluence

Authors: We used a laser pulse width of 7 ns and a fluence of 15 mJ/cm² throughout the study. We have added this information in the method section. The revised sentence reads as "*and a wavelength tunable (690 nm to 950 nm) OPO laser with a pulse width of 7ns, and a repetition rate of 10 Hz (Phocus Essential, Opotek, Inc., Carlsbad, CA, USA). The laser fluence used in the study is 15 mJ/cm².*" in Supplementary note 1.

4. The US acquisition time is 1 second and the bubbles are in the vasculature. Isn't there significant movement of the bubbles over this one second? Blood flow is quite fast.

Authors: Yes. There is significant movement of the bubbles over one second, but we did not need to track each bubble during the entire time. To extract flowing microbubble signals, we first used a spatial-temporal filter, which assumes that microbubble or blood signal movement is much faster than tissue movement to filter out the static tissue background. Then we sampled the remaining microbubble signals of the whole field-of-view in the frequency of 500 Hz. The trace of each moving microbubble within the field of view was constructed by the Hungarian tracking algorithm (The detailed description of bubble tracking is in Supplementary Note 3). Because of the fast blood velocity, microbubbles did not remain in the field of view for one second. Instead, some bubbles flow out of the field of view, and some could collapse during imaging. New microbubbles flowed into the field of view to produce new traces and

compensated for the lost ones because the microbubble densities in the bloodstream stayed approximately the same. What we have traced is each microbubble during certain frames, and we accumulated all traces to construct the super-resolution frames.

5. The authors mention that the algorithm identifies 92 of 100 bubbles. What happens to the other 8? They are lost to overlap or are out of depth or?

Authors: The bubbles are randomly dispersed. Statistically, a small portion of the bubbles are highly overlapped and beyond the resolution of the algorithm. For example, in the image (Figure 1a) below, we indicate the missing localization points with blue arrows. Figure 1b shows an example of two microbubbles separated by 25 μm . Since the distance of the microbubbles are beyond the limit that the Sparsity Constrained (SC) identification algorithm can resolve, only a single location will be recognized. This is the main source of the localization error. Figure 1c shows another example where microbubbles are separated by 64 μm , although microbubbles are still highly overlapped in the ultrasound image, the distance between them is within the SC identification limit, and the locations of both bubbles are identified. The situation will get more complicated when more than two microbubbles are overlapped. For example, from arrows 2 and 3 of Figure 1a, more than two microbubbles are overlapped, but only one position is recovered. Thus, in practice, the numbers of aggregated microbubbles, the amplitude of microbubble signals, and the distance between microbubbles are critical factors that cause the retrieving error.

Figure 1 | (a) Top: synthetic US image of 100 microbubbles (16 microbubbles/ mm^2). Bottom: the localization image recovered by the SC optimization. (b) Top: a Zoom-in US image of two microbubbles with a distance of 25 μm . Bottom: the localization image recovered by SC optimization. The results show that only one position is retrieved from two overlapped bubbles, indicating a

recovery error. (c) Top: a Zoom-in US image of two microbubbles with a distance of 64 μm . Bottom: the localization image recovered by SC optimization. In this case, both positions of the bubbles are successfully retrieved.

6. Figure 2a: One would think having a lower bubble density would lead to better counts. Lots of overlap here. Why does the PSF function behave so poorly?

Authors: We agree with the reviewer that PSF cross correlation method (PSF-CC) would lead to a better localization recovery efficiency in the case of lower bubble density than in higher bubble density because, statistically, the ratio of the aggregated microbubbles with low bubble densities is lower than the ratio in high-density bubbles. Using the synthetic data as an example, we assume the microbubble distribution is a random distribution (Gaussian distribution). To illustrate the effect, in Figure 2 below, we plot the distribution of microbubble separations in two different microbubble densities (16 and 104 microbubbles/ mm^2). From the distribution, the average separation distance of low-density is larger than that of high-density. The minimum distances between two microbubbles that PSF-CC and SC methods can distinguish are marked as dash lines in Figure 2. From the chart, we can see that in the low-density case (the red curve), the PSF-CC method can identify around one-third of the microbubbles (estimated by integrating the red curve in the right side of PSF-CC dash line), and in the high-density case, the PSF-CC method can barely identify any microbubbles. The error of the PSF-CC method in the low bubble density case is mainly from the overlapped microbubbles.

Figure 2 | Distance distribution of the microbubble separation to its nearest neighbor in the concentration of 16 (red) and 104 (blue) microbubbles/ mm^2 . The dashed lines represent the cut-off of the minimum separation in SC and PSF-CC methods.

The distorted PSF-like patterns shown in the main text Figure 2a are not a single PSF pattern but an overlay of multiple bubbles. In Figure 3 below, the arrows highlight examples of two and three bubble overlap cases in main text Figure 2a (where the bubble density = 16 microbubbles/ mm^2). The PSF-CC method is based on calculating the cross-correlation coefficient between the PSF pattern and the image. The method identifies the positions of microbubbles by finding the local maximum values of the cross-correlation map. It only compares the PSF pattern with the microbubble pattern pixel by pixel and finds the position with the “highest similarity” (the peak value in the correlation map). Figure 3b shows the identification of two overlapped microbubbles using PSF-CC methods. Only one peak value is found in the correlation map, and thus PSF-CC method can only identify one position. As mentioned in Point 5, the situation will get more complicated when more than two microbubbles are overlapped. Figure 3c

shows the identification of three overlapped microbubbles using the PSF-CC method. Although the correlation map shows several peaks, due to the low correlation value (less than 0.6, the threshold), no position is identified by the PSF-CC method. The poor performance in dealing with overlapped microbubbles is shown in the bottom panel of Figure 3a. The method identifies nothing or a single location in some multi-overlapped cases, indicated by the arrows.

Figure 3 | (a) Top: synthetic US image of 100 microbubbles (16 microbubbles/mm²). Bottom: the localization image recovered by the PSF-CC optimization. The arrows indicate the cases of overlapped microbubbles. (b) Top: US image of 2 microbubbles, and Bottom: normalized correlation map from PSF-CC. The red dot in the normalized map indicates the localization point recovered from two microbubbles by the PSF-CC method, indicating a retrieving error. (c) Top: US image of 3 microbubbles, and Bottom: normalized correlation map from PSF-CC. Because of the low correlation indices throughout the map (index threshold = 0.6), none of the points is retrieved, meaning all three bubbles are missed.

7. Figure 2c is important and illustrates the value of the work.

Authors: We appreciate the positive comment from the reviewer.

8. Some more explicit details on total imaging time are needed. The contributions of both signal acquisition and signal processing are needed.

Authors: Per the reviewer's suggestion, we have listed the total imaging time in table 1 below. Please note that the time reflects the current setup in the lab, not the fundamental limit. The data collection and signal processing were performed on a Dell workstation (Dell Precision 5820) with an Intel® Xeon® W-2155 Processor and 96 GB RAM. The PA imaging speed is limited by our laser, which has a pulse repetition rate and wavelength tuning rate of 10 Hz. In the lymph node imaging acquisition, each hybrid PA/ULM

sequence takes 1.8 sec (for 750 US B-mode frames and 3 PA frames), a total of 16 sequences were collected to reconstruct the 3D image. In kidney imaging, each hybrid PA/ULM sequence takes 1 sec (including 400 US B-mode frames and 2 PA frames). In the post-data processing, the processing time of lymph node imaging and kidney imaging for each frame is slightly different because the image size used in the processing is different. The US and PA image of the lymph node has 60x175 pixels, while the kidney image has 50x135 pixels. The pixel size of US and PA images is 100 μm . The final super-resolution ultrasound images of the lymph node and the kidney are 480x1400 pixels and 400x1080 pixels, respectively. The pixel size of super-resolution image is 12.5 μm . The PA processing time and ULM processing time for the lymph node images are 0.551 seconds per frame and 6.268 seconds per frame, respectively; for kidney images, the processing times are 0.503 seconds per frame and 5.857 seconds per frame, respectively.

In the post-image-processing, the bottleneck process is the optimization. However, it is worth noting that our Sparsity Constrained optimization uses fast shrinkage-thresholding (FISTA) optimization method and the property of BCCB matrix to efficiently compute vector operations and matrix-vector multiplications (mentioned in Supplementary Note 3). It thus can process the whole image while other methods are required to process the image patch by patch (it deals with only 7 by 7 pixels in each calculation). As a result, our SC optimization is faster than other methods and can prevent errors in stitching the images. To analyze the performance of optimization methods, we repeated the previously published convex-based optimization (CVX) and L1-homotopy (L1H) optimization methods on the same lymph node and kidney image data sets and compared their processing times with FISTA. As shown in Table 2, we list the processing time for these methods. It took 850.1 seconds and 65.9 seconds to process a frame of lymph node, while our FISTA method took only 6.268 seconds. Our FISTA optimization method is up to 137-fold and 10-fold faster than CVX and L1H, respectively. Further, the speed of FISTA optimization method can be improved. As shown in the discussion, GPU-accelerated FISTA is reported¹ and can be used to accelerate this process.

Table 1. the data acquisition and processing time of dual modal PA/ULM imaging.

	Data acquisition time (sec)			PAprocessing (sec/frame)		
	PA	US	Total	Image reconstruction	Spectrum unmixing	Total
Lymph node	0.300	1.500	1.800	0.014	0.537	0.551
Kidney	0.200	0.800	1.000	0.009	0.494	0.503

Table 1. (continued)

	ULM processing (sec/frame)				
	Image reconstruction	Clutter filtering	Motion correction	SC optimization	Total
Lymph node	0.014	0.026	0.028	6.200	6.268
Kidney	0.009	0.022	0.026	5.800	5.857

Table 2.the data processing time of the localization optimization algorithms.

	CVX(sec/frame)	L1H(sec/frame)	FISTA(sec/frame)
Lymph node	850.1	65.9	6.2
Kidney	545.7	42.3	5.8

CVX: convex; L1H:L1homotopy, FISTA fast shrinkage-thresholding algorithm

9. One of the key attributes of acoustic imaging is that it offers video frame rates relative to techniques like PET or CT. Some more comments on the slow frame rates and ideas to improve are needed in the discussion.

Authors: We agree with the reviewer that speed is one critical advantage of acoustic imaging over other medical imaging modalities. To bring the speed of dual-photoacoustic-ultrasound localization imaging to the speed of clinical ultrasound imaging is the primary motivation of this study. But we recognize this is not a simple task and will be accomplished step-by-step. We believe that the outcome of this study is the critical first step toward video rate dual-photoacoustic-ultrasound localization imaging.

Currently, PA imaging is not the limiting factor of video rate imaging. While we report a 10 Hz photoacoustic imaging (2-5 Hz of spectral-photoacoustic imaging). The frame rate here reflects the repetition rate of the laser we have. The wavelength tunable lasers with >100 Hz frame rate are commercially available. With a high frame rate laser, video-rate photoacoustic imaging (or spectral-photoacoustic imaging) is feasible.

The main limitation is from ultrasound localization imaging. Although our SC optimization using FISTA accelerated the optimization more than 100 times and 10 times compared to prior published CVX and LH1 optimization methods from Point 8, it is still far from the video rate. While others and we have greatly improved the speed of ultrasound localization imaging, much more work on the hardware and algorithm is needed to realize video-rate ultrasound localization imaging. In hardware, further improvement of ultrasound frame rate and data transfer speed, hardware beam-former such as FPGA beam-former, and high-performance graphic processing units will significantly release the current limitation. In software, GPU-based parallel processing, high-dimensional signal processing techniques such as deep learning algorithms may further improve the speed. These topics are currently being investigated in our lab and others. With the rapid evolution of the field, we believe that video-rate or faster ultrasound localization imaging can be realized in the foreseeable future, which we believe will further extend the impact of our dual-PA-ULM imaging in medical imaging applications. We have included these additional comments in the discussion section of our revised manuscript.

10. Might the value of interleave imaging be integrated for even further refinements? Some references and comments are needed.

Authors: We appreciate the reviewer recognized the value of interleaved imaging. As presented in our manuscript, interleaved imaging is especially important for *in vivo* imaging.

The interleaved recording reduces the burden of the spatial registration of the 3D image set of PA and ULM. We showed that inter-breathing cycle motion causes untraceable motion (elevational) artifacts, especially in a linear array system. The motion errors accumulated between the frames were recorded from different breathing cycles. In sequential scanning, a severe un-trackable inter-breathing cycle artifact and mechanical scanning error cause misalignment between the 3D image set of PA and ULM, even acquiring the images with respiratory gating. With interleaved recording and the fast ULM, we ensure each PA and ULM frame is recorded within one respiratory cycle, which greatly minimizes the effect of the un-trackable motion. It significantly reduces the effort for the co-registration of 3D images afterward. The lymph node image shown in figure 4e of the main text is an example where the co-registration of two image sets (3D PA images and 3D ULM images) enables the differentiation of the blood vessels within and outside the lymph node.

Looking forward, besides spatial registration, interleaved imaging is also crucial to time-lapse imaging. For example, it has been demonstrated that ULM can map deep-tissue blood velocity in unprecedented detail (main text Figure 6b). When it interleaves with PA imaging, it enables the unique imaging capability for investigating the correlation of localized velocity change and the diffusion (or localized concentration change) of imaging agents; or the correlation of localized velocity change with tissue oxygenation variation in time (main text Figure 6 h-i). Expanding from our current setup, high pulse energy nanosecond lasers with high repetition rates (10 kHz for single wavelength and 100 Hz for multiwavelength) are commercially available. Our method can be directly adopted when a high-repetition-rate laser is used. With a high-repetition-rate laser, it is possible to enable the frame-by-frame co-registration of microbubble tracking with PA images with a 1-microsecond temporal resolution.

Although the advantages of interleaved imaging are clear, examples of interleaved recording in clinical multimodality imaging are rare mainly because of the slow imaging speed and challenges in aligning the two imaging scanners, etc. The published interleaved reports focus on PA/US imaging and fast multimodality optical imaging, such as fluorescence imaging/optical coherence tomography (OCT) imaging. Per the reviewer's request, we have added an additional reference, "Ref 19", which reported a real-time interleaved PA/US imaging.

We added an extra paragraph in the discussion to elaborate further on the advantage of our interleaved imaging. The paragraph reads as follows: "Conventional dual-imaging conducts each 3D scan sequentially. Because of the un-trackable elevational motions in an ultrasound linear array system, the motion errors accumulate between the frames recorded from different breathing cycles. The severe un-trackable inter-cycle artifacts and mechanical scanning errors can cause misalignment between 3D imaging sets, complicating the image registration process. With an interleaved recording and the fast SC-ULM, we ensure each PA and ULM frame is recorded within one breathing cycle, which greatly minimizes the effect of un-trackable motion. Our results show that interleaved imaging significantly reduces the burden of the spatial registration of the 3D PA and ULM imaging sets. The interleaved recording is also crucial to 2D time-lapse imaging. The alternative recording of PA and ULM images over time minimizes the un-trackable motions between two images and improves the temporal and spatial registration of the time-lapse imaging sets."

.....

Reviewer #2:**Summary:**

This work proposes a dual PA/ULM technique for vascular imaging and physiological imaging computed at 2 second intervals. The SRUS method of Sparsity is used to accelerate saturation of the vessel images. The methods are demonstrated *in vivo*. The novelty of the paper seems to be the use of PA interleaved with ULM. Only one image is shown as PA/ULM, that of the dye labeled mouse lymph node “with nearby microvasculature”. The kidney work is not displayed as PA/ULM, perhaps because the information does not fuse. The case for a dual mode of interleaved operation of PA and ULM, versus a sequential one, is not clear. The authors state that the Sparsity based method for ULM is novel, however, the case is not well founded. The conclusion of the paper, mainly resting on “development of a sparsity-constrained” ULM method with computational speedup does not stand. Overall, the paper is well organized and well written. There were some grammatical errors (not noted), but it did not impede the review.

Authors: We thank the reviewer’s critique and appreciate the reviewer’s recognition of our novelty in interleaving PA/ULM and the quality of the paper. We want to point out that the primary goal of this research is to develop a dual PA/ULM imaging technique to harness complementary imaging capabilities from both modalities. To the best of our knowledge, this work is the first demonstration of *in vivo* dual PA/ULM imaging. While the idea of combining two imaging techniques seems simple since ultrasound/photoacoustic imaging has been widely deployed, the inherently long data acquisition time required for super-resolution ultrasound and subsequent challenges of motion artifacts *in vivo* make the tasks challenging.

Inspired by the existing sparsity methods (as cited in reference 39-42 in main text), we developed a fast ULM imaging to pair with PA imaging by overcoming the main hurdles. Although we are inspired by the prior methods, our sparsity-constrained method is not identical to any of the cited methods. For example, in reference 39 (the SUSHI method), SUSHI uses sparsity recovery to solve the statistic independence of flowing microbubble signal fluctuation in a correlation domain. In contrast, we used the sparsity recovery to retrieve the positions of overlapped microbubbles over frames directly. Although both methods use a fast iterative shrinkage-thresholding algorithm (FISTA) to solve the optimization problem, SUSHI conducts the optimization in the 2D Fourier domain, and our method solves the optimization in the time domain. Thus, the input and output of our method are very different from SUSHI. Compared to other methods, such as reference 41 and reference 42, two compressed sensing methods are used to directly retrieve the position of the microbubbles, which is similar to what we did with the sparsity-constrained method. But these published methods used much slower convex (CVX) and l-1 homotopy (L1H) optimizations, and they required image stitching. Differently, our process uses faster optimization and stitching *free*.

In our manuscript, we showcased two imaging scenarios to represent the two separate advantages of the developed imaging protocols. The first lymph node imaging is to highlight the resolution and its capability in spatially co-registering the images. On the other hand, the mouse kidney imaging mainly shows the advantage of the high-speed ULM that is robust against motion artifacts; thus, we did not provide the dual contrast image in the first submission. But we fully understand the reviewer’s concern about the missing dual image in the second example. Here in the revised manuscript, we present a new figure (main text Figure 6) to demonstrate the co-registered images of super-resolution ULM blood velocity image and its corresponding PA oxygenated hemoglobin image in a mouse kidney. In the new study, we conducted an *in vivo* oxygen-challenging experiment in mice. During the imaging, the mouse inhaled 100%, 3%, and 20% of oxygen for 50 seconds alternatively. We showed co-registered time-lapse images of super-resolved

renal microvascular velocity distribution recorded by our SC-ULM and the map of the kidney tissue oxygenation level recorded by label-free PA imaging. This example highlights the necessity of interleaved recording rather than sequential recording. Because of the interleaved recording, we can spatially and temporally co-register dual PA and ULM kidney images during different time period; as a result, we are able to identify the relation of renal blood flow to hemoglobin oxygenation at the same location over 150 seconds (main text Figure 6 h-l). Both examples demonstrate the significance of PA/ULM interleaved imaging.

Further, we added a paragraph to elaborate better on the advantages and significance of PA/ULM interleaved imaging. PA/ULM interleaved imaging can minimize the motion artifacts and enable better temporal and spatial co-registration in 3D scanning and time-lapse imaging. The new paragraph reads as, “Conventional dual-imaging conducts each 3D scan sequentially. Because of the un-trackable elevational motions in an ultrasound linear array system, the motion errors accumulate between the frames recorded from different breathing cycles. The severe un-trackable inter-cycle artifacts and mechanical scanning errors can cause misalignment between 3D imaging sets, complicating the image registration process. With an interleaved recording and the fast SC-ULM, we ensure each PA and ULM frame is recorded within one breathing cycle, which greatly minimizes the effect of un-trackable motion. Our results show that interleaved imaging significantly reduces the burden of the spatial registration of the 3D PA and ULM imaging sets. The interleaved recording is also crucial to 2D time-lapse imaging. The alternative recording of PA and ULM images over time minimizes the un-trackable motions between two images and improves the temporal and spatial registration of the time-lapse imaging sets.”

Regarding the novelty of our study, we want to reiterate that while our sparsity-constrained ULM method is part of our novel approach toward the goal, the primary novelty lies in overcoming the overall technical difficulty and demonstrating the first *in vivo* PA/ULM imaging. In our study, we have shown that this combined imaging technique can image living mice to generate multi-dimensional informative images for the first time.

Finally, we appreciate the reviewer’s comments about our conclusion section, which seems to focus only on the acceleration of the ULM. We have rewritten the conclusion section to better reflect the overall novelty and the accomplishment of this study. The new conclusion reads as “In conclusion, we demonstrate the first dual PA/super-resolution US imaging and its applications in scenarios that are challenging to image using other modalities. We show that this imaging technique can generate dual contrast images to reveal a dye-labeled lymph node as well as micron sized blood vessels within and near the lymph node. We also demonstrate that the imaging maps the renal oxygenation and the blood flow of the microvasculature in a mouse kidney over 2 minutes. We expect this dual imaging modality will promote a broad interest in imaging applications for cancer diagnosis or other imaging applications in neuroscience, nephrology, and immunology.”

Major Comments:

The authors claim:

However, SUSHI requires a high frame rate to resolve fine vessel structures, and the resolution is lower than localization based super-resolution methods.

And

However, the proposed compressed sensing methods for localization used patch-stitch image processing, which may lose computational speed and accuracy.

The reference cited (38) shows an algorithm (Algorithm 1) which is nearly identical to the one the authors provide in the supplement. Would the authors please describe where there are meaningful differences. While this reference (38) used a high frame rate, and benefited from that, it is not stated as required for the method. Many high quality ULM images have been made using frame rates of hundreds of Hz. There appears to be comparable accuracy of the SUSHI with FISTA method (38) with super localization – shown in Figure 3 of that work. The simulation methods showing the SC method recovers more MB is not novel.

Authors: We thank reviewer’s comments on the comparison of our methods with the SUSHI method. There is no doubt that SUSHI is a powerful technique that pioneers fast super-resolution ultrasound imaging. However, the principle of SUSHI and our method (SC-ULM) is fundamentally different, although we both used FISTA for optimization. To reconstruct a super-resolution ultrasound image, the general steps for SUSHI and our SC-ULM require to include developing a forward model and then solving the inverse problem. In the following, we provide a side-by-side comparison of SUSHI and SC-ULM in terms of constructing the forward model and solving the inverse problem.

Forward model:

For SUSHI and SC-ULM, the ultrasound signal $f(x, z, t)$ is composed of the blood signal $b(x, z, t)$ (or microbubble signal) and contaminated by the tissue clutter $c(x, z, t)$ and an additive noise component $w(x, z, t)$:

$$f(x, z, t) = c(x, z, t) + b(x, z, t) + w(x, z, t), \quad (1)$$

The first process for SUSHI and SC-ULM is to extract blood signals from ultrasound signals. SC-ULM applied a spatiotemporal singular value decomposition (SVD) clutter filter to remove the tissue signal, but SUSHI used a sixth-order Butterworth filter to separate the tissue and blood signals. SVD clutter filter explores different features of tissue and blood motion in terms of spatiotemporal coherence, while Butterworth filter only considers frequency difference between tissue and blood signal. The comparison of SVD clutter filter and Butterworth filter is well documented in². In addition, SUSHI used Doppler bands in the frequency domain to separate blood signals with different velocities.

After decluttering the signal, in SC-ULM, we assume the linear shift-invariant of the ultrasound images. The decluttered signal $b(x, z, t)$ is the convolution of the point spread function (PSF) of the ultrasound system $h(x, z)$ and the distribution of microbubble signal $s(x, z, t)$:

$$b(x, z, t) = h(x, z) * s(x, z, t). \quad (2)$$

In the discretization form, we can assume the size of the ultrasound image is $M \times M$ pixels. To achieve super-resolution, we discretize the image by creating a high-resolution grid with the image size of $N \times N$ pixels.

$$b[m, n, t] = \sum_{i=0}^{N-1} \sum_{j=0}^{N-1} h[mP - i, nP - j]s[i, j, t], \quad (3)$$

where P is the downsampling factor, $N = MP$.

Rewriting into the matrix form:

$$\mathbf{b}(t) = \mathbf{D}\mathbf{H}\mathbf{s}(t), \quad (4)$$

where $\mathbf{b}(t) \in \mathbb{C}^{M^2 \times 1}$ is vectorized blood signal, \mathbf{D} is the downsampling operator, $\mathbf{H} \in \mathbb{C}^{N^2 \times N^2}$ is BCCB matrix related to the PSF of the ultrasound system, $\mathbf{s}(t) \in \mathbb{C}^{N^2 \times 1}$ is the microbubble distribution on high resolution grid. Equation (4) is the forward model of SC-ULM.

Very different from our methods, SUSHI calculates the sparsity in the correlation domain and assumes the statistic independence of the temporal signal fluctuation of different blood vessels. It calculates the cross-correlation of blood signal with the time lag τ , and gets:

$$\mathbf{g}(\tau) = \mathbf{Cov}(\mathbf{b}(t), \mathbf{b}(t + \tau)) = \mathbf{D}\mathbf{H}_2\mathbf{u}(\tau), \quad (5)$$

where $\mathbf{H}_2 = (H_2[i, j]) = (|H[i, j]|^2)$, means every element in \mathbf{H}_2 is the square of the absolute value of every element in \mathbf{H} , $\mathbf{u}(\tau)$ represents the autocorrelation of all pixels on high resolution grid for a prechosen time-lag τ .

Using Fourier transform,

$$\mathbf{G}(\tau) = \mathbf{H}_0(\mathbf{F}_M \otimes \mathbf{F}_M)\mathbf{u}(\tau), \quad (6)$$

where $\mathbf{H}_0 = \text{diag}\{\tilde{H}[0,0], \dots, \tilde{H}[M-1, M-1]\}$ is $M^2 \times M^2$ diagonal matrix. $\tilde{H}[k_m, k_n]$ is the $M \times M$ 2-D DFT (Discrete Fourier transform) of the $M \times M$ squared, absolute value PSF $|h(mP, nP)|^2$. \mathbf{F}_M denotes a partial $M \times N$ DFT matrix and \otimes stands for the Kronecker product. (6) is the forward model of SUSHI.

Inverse Problem:

After constructing the forward models in a matrix-vector form, $\mathbf{y} = \mathbf{A}\mathbf{x}$, both SC-ULM and SHUSHI require to solve the inverse problem through optimization to calculate \mathbf{x} . While the FISTA algorithm, as listed in Algorithm 1 (Equation 7 below), was used for accelerating the optimization process, the input and output of both methods are very different.

$$\mathbf{x} = \arg \min_{\mathbf{x}} \frac{1}{2} \|\mathbf{y} - \mathbf{A}\mathbf{x}\|^2 + \lambda \|\mathbf{x}\|_1. \quad (7)$$

In SUSHI, the input $\mathbf{y} = \mathbf{G}(\tau)$, $\mathbf{A} = \mathbf{H}_0(\mathbf{F}_M \otimes \mathbf{F}_M)$. While in SC-ULM, the input $\mathbf{y} = \mathbf{b}(t)$ and $\mathbf{A} = \mathbf{D}\mathbf{H}$.

In SUSHI, after solving the optimization, the solution (output) is the super-resolved image. In SC-ULM, the optimization is solved frames by frames (over time); and the solution (output) of the optimization represents the microbubbles' locations at a time, t .

We do agree with the reviewer that "While this reference (38) used a high frame rate and benefited from that, it is not stated as required for the method. Many high-quality ULM images have been made using frame rates of hundreds of Hz." To prevent confusion, we have deleted the sentence "However, SUSHI requires a high frame rate to resolve fine vessel structures ..." from our introduction.

Regarding to the simulation methods, we agree with the reviewer that our simulation method is not novel. However, the novelty of the simulation method is not what we would like to claim, instead, the reason that we brought up the similar analysis methods of the simulation like the methods used in the reference 42 is to make a fair comparison of the performance of different localization methods.

Detailed Comments

Manuscript:

1. p. 15 The section on performance *in vivo* being different is weak. The biological mechanisms leading to great implications on performance in realistic environments appear to be poorly understood. Phase changing nanodroplets are not the solution to "unbalanced vessel sizes". The droplets may extravasate and tracing the vasculature may be more problematic.

Authors: We respectfully disagree with the reviewer's comments on "*The biological mechanisms leading to great implications on performance in realistic environments appear to be poorly understood.*" We did not think our method suffered from *in vivo* imaging compared to other methods. It is expected that there is a difference between the theoretical performance and the *in vivo* performance. The reason includes motion artifacts and uneven contrast agent distribution caused by the blood flow variations. The fast sparsity method significantly reduces the motion artifacts. The first paragraph of page 15 of our original main text discusses the potential negative effect of blood flow variation in renal vasculature compared to the vasculature in the hindlimb. Please note the blood flow variation in different sizes of the blood vessels are well documented³, and its consequence to the super-resolution ultrasound imaging has been discussed. Here we would like to directly quote a discussion from SUSHI⁴ "*SUSHI shares an inherent physical limitation of all CEUS imaging methods with short acquisition times, even when high US contrast agent concentrations are used: certain vessels with very low flow velocities might not contain microbubbles during the imaging interval. This is also true for both SUSHI and CEUS imaging methods with lower CEUS concentrations and longer acquisition times, no matter what processing method is applied.*". Below we quote another discussion from the ULM paper that we referenced⁵, which states that "*....Ultrasound Localization Microscopy was recently introduced to overcome this limit and relies on subwavelength localization and tracking of microbubbles injected in the blood circulation. Yet, as microbubbles follow blood flow, long acquisition times are required to detect them in the smallest vessels, leading to long reconstruction of the microvasculature.*" Concluding from the literature and our own experience, the non-uniform blood flow and the distribution of microbubbles is one of the critical reasons that lead to the organ-dependent imaging performance in ULM, although it may not be the sole cause.

We proposed using nanodroplets as a potential solution to reduce the effect of the non-uniform contrast agent dispersion, but we did not claim it is the ultimate solution to bridge the gap between *in vivo* imaging and theoretical study. The reasons we think nanodroplets can reduce the effect of vascular size-dependent non-uniform ULM signals are the following. Blood flow dynamic studies have shown that, compared to the micron-size nanoparticle, there is a very different margination and adhesion dynamic in nano-size contrast agents⁶. Although 200 nm – 500 nm droplets may extravasate after injecting into blood vessels, the extravasation of the nanoparticle in this size range is not a rapid process⁷. It will take hours to induce a noticeable effect, especially in the intact non-leaking vasculature. The recent *in vivo* imaging study shows that the extravasation does not significantly affect the localization⁸.

We have conducted additional experiments to investigate the signal uniformity improvement using nanodroplets. In Figure 6 below, we show that by injecting the same concentration of nanodroplets and microbubbles, the distribution of nanodroplets is more uniform than microbubbles.

Figure 6 | Contrast enhanced ultrasound images show the distribution of (a) phase changing nanodroplets and (b) microbubbles in mouse cerebral vasculature. The images were produced from one frame of raw contrast enhanced ultrasound image and processed by SVD decluttering to remove the static tissue signal. The boxes highlight the areas where microbubble signals overlap but the nanodroplet signals do not. The bottom two images are raw Power Doppler images of the same mouse brain with (c) phase changing nanodroplets and (d) microbubbles as contrast agents. The boxes highlight the areas where the microbubbles cause the signal saturation due to the microbubble aggregation but the nanodroplet signals are more uniform across the area, greatly reducing the signal saturation.

2. p. 15 Nanodroplets are not a solution to MB dosage concerns, they are not yet FDA approved for instance.

Authors: We mentioned nanodroplets but didn't intend to promote them as an ultimate solution for microbubble dosage concerns. Our intention is to consider nanodroplets as a potential alternative approach to elongate the imaging time without increasing the contrast agent dosage. Studies have shown

that the blood half-life of the lipid perfluoro-nanodroplets is about 3 hours in the circulation⁹. Compared to ~5 minutes half-life of the microbubbles¹⁰, nanodroplets did offer much longer imaging time.

We agree with the reviewer that nanodroplets have not yet been approved by FDA. However, we would like to point out that it is possible to prepare acoustic activatable nanodroplets from FDA-approved materials¹¹. This approach could significantly shorten the FDA approval in the future. In addition, nanodroplets have been an active study in ultrasound contrast agent development and have demonstrated a great degree of success in pre-clinical studies. We believe some of them will be translated in the near future. Knowing that nanodroplets may benefit the imaging time of ULM could be another big push for more efforts in clinical translation. However, we would like to emphasize that we agree with the reviewer that there are other approaches that can facilitate the dosage issue, such as faster ULM imaging, safer microbubbles that increase the allowable dosage, or the new coating layer to elongate the circulation of microbubbles. To prevent confusion, we have deleted the discussion of nanodroplets.

3. Various passages use present tense when past is more natural.

Authors: We thank the reviewer for pointing out the grammar errors. We have corrected them.

4. References (many) are incomplete – lacking journal name.

Authors: We thank the reviewer for pointing out this. We have corrected the references with missing journal names.

Supplementary information:

1. Please provide a brief description of the algorithms for motion correction and tracking with references.

Authors: We conducted the motion correction with a published protocol¹² and described the protocol in Supplementary Note 8. In our imaging, we observed that respiratory and cardiac motions are the two primary sources of motion artifacts. Respiratory motions are dominant and corrected for both lymph node imaging and kidney imaging. In kidney imaging, the additional cardiac pulsation is severe enough to degrade the imaging quality; thus, it needs to be corrected. Specifically, we first calculated the frame-to-frame normalized cross-correlation of the recorded ultrasound images for identifying the out-of-plane respiratory motions. The large out-of-plane respiratory motion resulted in a correlation index drop. The ultrasound frames with a low index below the threshold (index < 0.99) were removed to reject unattractable out-of-plane motions. The rest in-plane motions were corrected using a rigid transformation model with gradient descent optimization. It assumes that each remaining ultrasound frame is a rotated and translated replica of the reference frame.

The shifting of the tracking algorithm is based on the Hungarian method and is described in Supplementary Note 3. We acquired microbubble positions from all the image frames first, then moving microbubbles were tracked with adjacent frames to discard uncorrected localized positions using a particle tracking algorithm (simpletracker.m available on Mathworks). Briefly, the algorithm calculates all distances between all microbubbles in the current frame and all microbubbles in the next frame. Then the algorithm minimizes the total distance and finds the optimal pairs of microbubbles in adjacent frames.

Applying the algorithm to all frames can give us collections of microbubble tracks. The final super-resolved image was reconstructed by superimposing all microbubble traces. The super-resolved velocity map can also be accessed by calculating the velocities of each microbubble using the distance of trajectory and the time interval between frames.

2. Various passages use present tense when past is more natural.

Authors: We thank the reviewer for pointing out the grammar errors. We have corrected them.

3. Please provide references for linear spectral unmixing.

Authors: Per the reviewer's suggestion, we have added the reference, [ref 10] in the supplementary information, and a short description for linear spectral unmixing.

Briefly, the measured PA signal after optical fluence normalization is a linear combination of the spectral signatures of different chromophores (i.e. hemoglobin, deoxyhemoglobin and an exogenous contrast agent) with the relative concentration of each chromophore:

$$P'(\lambda_i, x, y) = \varepsilon_{HbR}(\lambda_i)C_{HbR}(x, y) + \varepsilon_{HbO_2}(\lambda_i)C_{HbO_2}(x, y) + \varepsilon_{exo}(\lambda_i)C_{exo}(x, y),$$

where $P'(\lambda_i, x, y)$ is the photoacoustic signal recorded at a wavelength λ_i after optical fluence normalization, $\varepsilon_{HbR}(\lambda_i)$, $\varepsilon_{HbO_2}(\lambda_i)$ and $\varepsilon_{exo}(\lambda_i)$ are known as molar extinction coefficients. $C_{HbR}(x, y)$, $C_{HbO_2}(x, y)$ and $C_{exo}(x, y)$ are the molar concentrations of different chromophores. By measuring PA signal at different wavelengths and using the spectrum of the known molar extinction coefficients, we can use linear least square methods to extract the molar concentrations of different chromophores:

$$\begin{bmatrix} C_{HbR}(x, y) \\ C_{HbO_2}(x, y) \\ C_{exo}(x, y) \end{bmatrix} = (\boldsymbol{\varepsilon}^T \boldsymbol{\varepsilon})^{-1} \boldsymbol{\varepsilon}^T \mathbf{P}',$$

where $\mathbf{P}' = \begin{bmatrix} P'(\lambda_1, x, y) \\ P'(\lambda_2, x, y) \\ \vdots \\ P'(\lambda_N, x, y) \end{bmatrix}$ and $\boldsymbol{\varepsilon} = \begin{bmatrix} \varepsilon_{HbR}(\lambda_1) & \varepsilon_{HbO_2}(\lambda_1) & \varepsilon_{exo}(\lambda_1) \\ \varepsilon_{HbR}(\lambda_2) & \varepsilon_{HbO_2}(\lambda_2) & \varepsilon_{exo}(\lambda_2) \\ \vdots & \vdots & \vdots \\ \varepsilon_{HbR}(\lambda_N) & \varepsilon_{HbO_2}(\lambda_N) & \varepsilon_{exo}(\lambda_N) \end{bmatrix}$.

In the experiment, the pulse energy is measured by the power meter, and the optical fluence is calculated by the pulse energy over the illuminated surface area. For the lymph node experiment, the exogenous contrast agent is ICG. For the kidney experiment, there is no exogenous contrast agent used, therefore $C_{exo}(x, y)$ is set to 0.

4. Figure S2 – lower row, right image – missing the red curve or not visible, it may lie near y-axis.

Authors: We thank the reviewer for pointing out the missing curve in Figure S2. In Fig S2c, the top figure shows the computation time of CVX (origin curve) is much longer than other methods. To show the comparison of computation time of different methods at the low MB density, we zoomed in the curves into red box. However, we can see in revision figure below, there is no sampling point for CVX (origin curve) in red box region. Therefore, we did not show CVX curve in original figure. We added additional caption in Figure 2 to explain it.

5. References 6, 7, 9, 11 are incomplete – lacking journal name.

Authors: We thank the reviewer for pointing this out. We have corrected these references.

Reference

1. Xu Q, Yang D, Tan J, Sawatzky A, Anastasio MA. Accelerated fast iterative shrinkage thresholding algorithms for sparsity-regularized cone-beam CT image reconstruction. *Medical physics* **43**, 1849-1872 (2016).
2. Demene C, *et al.* Spatiotemporal clutter filtering of ultrafast ultrasound data highly increases Doppler and fUltrasound sensitivity. *IEEE Transactions on Medical Imaging* **34**, 2271-2285 (2015).
3. Ivanov KP, Kalinina MK, Levkovich YI. Blood flow velocity in capillaries of brain and muscles and its physiological significance. *Microvascular Research* **22**, 143-155 (1981).
4. Bar-Zion A, Solomon O, Tremblay-Darveau C, Adam D, Eldar YC. SUSHI: sparsity-based ultrasound super-resolution hemodynamic imaging. *IEEE Transactions on Ultrasonics, Ferroelectrics, and Frequency Control* **65**, 2365-2380 (2018).
5. Hingot V, Errico C, Heiles B, Rahal L, Tanter M, Couture O. Microvascular flow dictates the compromise between spatial resolution and acquisition time in Ultrasound Localization Microscopy. *Scientific Reports* **9**, 2456 (2019).

6. Cooley M, Sarode A, Hoore M, Fedosov DA, Mitragotri S, Gupta AS. Influence of particle size and shape on their margination and wall-adhesion: implications in drug delivery vehicle design across nano-to-micro scale. *Nanoscale* **10**, 15350-15364 (2018).
7. Popović Z, *et al.* A nanoparticle size series for in vivo fluorescence imaging. *Angewandte Chemie* **122**, 8831-8834 (2010).
8. Riemer K, *et al.* Fast and selective super-resolution ultrasound in vivo with sono-switchable nanodroplets. *arXiv preprint arXiv:220304263*, (2022).
9. Bérard C, *et al.* Perfluorocarbon nanodroplets as potential nanocarriers for brain delivery assisted by focused ultrasound-mediated blood–brain barrier disruption. *Pharmaceutics* **14**, 1498 (2022).
10. Stride E, *et al.* Microbubble agents: new directions. *Ultrasound in medicine & biology* **46**, 1326-1343 (2020).
11. Sheeran PS, Yoo K, Williams R, Yin M, Foster FS, Burns PN. More than bubbles: creating phase-shift droplets from commercially available ultrasound contrast agents. *Ultrasound in medicine & biology* **43**, 531-540 (2017).
12. Foiret J, Zhang H, Ilovitsh T, Mahakian L, Tam S, Ferrara KW. Ultrasound localization microscopy to image and assess microvasculature in a rat kidney. *Scientific Reports* **7**, 13662 (2017).

REVIEWERS' COMMENTS

Reviewer #1 (Remarks to the Author):

Reviewer #1 here: The authors have largely addressed my concerns in the letter but more of the key details found in the response letter need to be integrated into the main text or SI so all readers can benefit. Points 2, 4, 5, and 6 need more of these details (e.g., letter Figure 2) in the text to be published. Also the paragraph on interleave does not have citations to prior work. These are needed: <https://www.nature.com/articles/s41467-021-20947-5> and <https://opg.optica.org/abstract.cfm?uri=ol-47-14-3503>.

Reviewer #2 (Remarks to the Author):

The authors have addressed all of my comments thoroughly. I congratulate them on this demonstration of a powerful new dual imaging strategy!

Response to Reviewers' Comments on manuscript **NCOMMS-22-36128A** entitled “**Hybrid Photoacoustic and Fast Super Resolution Ultrasound Imaging**” by Shensheng Zhao, Jonathan Hartanto, Ritin Joseph, Cheng-Hsun Wu, Yang Zhao, Yun-Sheng Chen*

Reviewer #1:

The authors have largely addressed my concerns in the letter but more of the key details found in the response letter need to be integrated into the main text or SI so all readers can benefit. Points 2, 4, 5, and 6 need more of these details (e.g., letter Figure 2) in the text to be published. Also the paragraph on interleave does not have citations to prior work. These are needed:

<https://www.nature.com/articles/s41467-021-20947-5> and <https://opg.optica.org/abstract.cfm?uri=ol-47-14-3503>.

Authors: We appreciate the positive comments from the reviewer! Per the reviewer's suggestion, we cited the two prior papers on interleaving (Ref 19 and 20). We also integrated the responses for Points 2,4,5 and 6 into either the main text or the SI.

For Point 2, we added the information “*each wavelength takes 0.1 seconds*” to “The interleaved imaging sequence for a dual PA/fast ULM imaging system” of the Results Section.

For Point 4, we added the details about the tracking algorithm in “Sparsity constrained optimization and PSF cross correlation” in the Method section.

For Point 5, we add the information in the second paragraph of the Results section. We also added the discussion of the localization limitation of SC and PSF-CC algorithm into Supplementary Note 1.

For Point 6, we put the figure on the microbubble distance distribution under different concentrations into the supplementary information (Supplementary Figure S3). We also included the discussion about the localization performance with different microbubble densities in Supplementary Note 2.

Reviewer #2:

The authors have addressed all of my comments thoroughly. I congratulate them on this demonstration of a powerful new dual imaging strategy!

Authors: We appreciate the positive comments from the reviewer!